# ASSESSING THE INTERPRETABILITY OF PROGRAMMATIC POLICIES USING LARGE LANGUAGE MODELS

## ABSTRACT

Programmatic representations of policies for solving sequential decision-making problems often carry the promise of interpretability. However, previous work on programmatic policies has only presented anecdotal evidence of policy interpretability. The lack of systematic evaluations of policy interpretability can be attributed to user studies being time-consuming and costly. In this paper, we introduce the LLM-based INTerpretability (LINT) score, a simple and cost-effective metric that uses large-language models (LLMs) to assess the interpretability of programmatic policies. To compute the LINT score of a policy, an LLM generates a natural language description of the policy's behavior. This description is then passed to a second LLM, which attempts to reconstruct the policy from the natural language description. The LINT score measures the behavioral similarity between the original and reconstructed policies. We hypothesized that the LINT score of programmatic policies correlates with their actual interpretability, and evaluated this hypothesis in the domains of MicroRTS and Karel the Robot. Our evaluation relied on a technique from the static obfuscation literature and a user study, where people with various levels of programming proficiency evaluated the interpretability of the programmatic policies. The results of our experiments support our hypothesis. Specifically, the LINT score decreases as the level of obfuscation of the policies increases. The user study showed that LINT can correctly distinguish the "degree of interpretability" of programmatic policies generated by the existing algorithms. Our results suggest that LINT can be a helpful tool for advancing the research on interpretability of programmatic policies.

## 1 INTRODUCTION

There is a growing interest in the use of programmatic representations of policies to solve sequential decision-making problems (Verma et al., 2018; Trivedi et al., 2021; Mariño et al., 2019; Qiu & Zhu, 2022; Liu et al., 2023). This interest is justified because one can provide a strong inductive bias to the learning process through the domain-specific language that defines the space of policies. This bias has been shown to allow programmatic policies to generalize more easily to unseen settings (Inala et al., 2020) and to make them more amenable to formal verification (Bastani et al., 2018).

Another feature of programmatic policies that is often emphasized in previous work is interpretability. However, no systematic studies have been performed that assessed the interpretability of these policies. A common method is to present specific programs and claim that they are interpretable (Verma et al., 2018; Trivedi et al., 2021; Aleixo & Lelis, 2023). The scarcity of comprehensive evaluations could be attributed to the fact that such studies are time consuming and costly. This lack of a thorough analysis hinders our understanding of what makes a programmatic policy interpretable. For example, neural networks can be viewed as programs written in a domain-specific language that allows the addition of layers and nodes to the neural architecture. Clearly, the programmatic framing of policies does not guarantee interpretability. Moreover, it is unreasonable to assume that the domain-specific language that defines the policy space is the only determinant factor for interpretability. This is because different programs written in the same language could have different levels of interpretability. So, how can we make progress on interpretable programmatic policy generation beyond just anecdotal claims and without time-consuming and costly human studies?

In this paper, we introduce a simple and cost-effective methodology to assess program interpretability and demonstrate its application to programmatic policies. Our methodology uses large language models (LLMs) to assign an interpretability score to a programmatic policy. We call this score the LLM-based INTerpretability (LINT) score. In our methodology, we use an LLM to generate a natural-language explanation of the behavior of the policy. This explanation is given as input to another LLM, which is asked to reconstruct the program described in the explanation. A third LLM verifies that the explanation is in natural language and is about the behavior of the policy. Once we obtain a reconstructed policy, the LINT score is the value of a metric that compares the behavior of the original and reconstructed policies, for example, the action agreement of the policies in a set of states. The underlying assumption of the LINT score is that a policy is interpretable if one can reconstruct it from a natural language description of its behavior.

We hypothesize that the LINT score of programmatic policies correlates with their actual interpretability. We further hypothesize that the programmatic policies previously considered interpretable vary in their degree of interpretability, and that the LINT score can capture this variation. We evaluated our hypothesis in the domains of MicroRTS (Ontañón et al., 2018), a real-time strategy game, and Karel the Robot (Pattis, 1994). Previous work presented programmatic policies for these two domains that were considered interpretable (Trivedi et al., 2021; Aleixo & Lelis, 2023). Our evaluation is divided into two parts. The first part is based on methods from the program obfuscation literature (Collberg & Nagra, 2009). Obfuscated programs are designed to be non-interpretable, and some obfuscation techniques allow us to construct programs with different levels of obfuscation. The assumption is that obfuscation can be used as a proxy for interpretability. The second part presents two user studies ($n = 60$ for MicroRTS and $n = 33$ for Karel) in which people with programming experience evaluated the interpretability of a set of programmatic policies.

The results of our experiments support our hypotheses. In both MicroRTS and Karel the Robot, the LINT score decreases as policy obfuscation increases. The user study further demonstrated that the LINT score effectively distinguishes the degree of interpretability of policies current algorithms generate for MicroRTS and Karel. However, the results also showed that the LINT score may struggle to accurately rank policies when the interpretability gap between programs is "too small". For policies with smaller gaps, the ground truth ranking from the user study may be unstable, and could vary with a larger participant pool or participants with different backgrounds. We conjecture that the challenges LINT faces with such unstable rankings is similar to how humans would find challenging to differenciate similarly interpretable policies. Our results also support the hypothesis that some previously considered interpretable policies exhibit varying degrees of interpretability, as both study participants and LINT flagged certain policies as less interpretable than others.

These findings demonstrate the potential of the LINT score as a tool for advancing research on the interpretability of programmatic policies. They also suggest avenues for future studies such as the development of interpretability metrics that can be used to select interpretable programs to present to users, when multiple options are available, as in program synthesis tasks (Singh & Gulwani, 2015).

## 2 Problem Definition

We are interested in finding interpretable solutions to sequential decision-making problems, which we formalize as Markov decision processes (MDPs) $(S, A, p, r, \mu, \gamma)$. $S$ is the set of states and $A$ is the set of actions. $p(s_{t+1}|s_t, a_t)$ is the transition model, which returns the probability of observing $s_{t+1}$ given that the agent is in $s_t$ and applies the action $a_t$ at the time step $t$. The agent observes a reward value of $R_{t+1}$ when moving from $s_t$ to $s_{t+1}$. The reward value is returned by the function $r$. $\mu$ is the distribution of the initial states of the MDP; states sampled from $\mu$ are denoted $s_0$. $\gamma$ in $[0, 1]$ is the discount factor. A solution to an MDP is a policy $\pi$, which is a function that receives a state $s$ and returns the action that the agent should take at $s$. The objective is to learn a policy $\pi$ that maximizes the expected sum of discounted rewards for $\pi$ starting in $s_0$: $\mathbb{E}_{\pi,p,\mu}[\sum_{k=0}^{\infty} \gamma^k R_{k+1}]$.

In addition to learning policies that maximize the expected sum of discounted rewards, we want to learn policies that are human-interpretable. Interpretability offers several advantages, including the ability for users to manually modify learned policies and the ability to formally verify policies (Bastani et al., 2018). We try to obtain interpretable policies by using programmatic representations (Trivedi et al., 2021). We search in the space of programs defined by a domain-specific language for a program $\pi$ that encodes an optimal policy. The task of learning a policy is equivalent

to a discrete search problem over the space of programs that the DSL accepts. The set of programs a DSL accepts is defined through a context-free grammar $(M, N, R, I)$, where $M$, $N$, $R$, and $I$ are the sets of non-terminals, terminals, the production rules, and the grammar's initial symbol. Figure 1 shows a DSL, where $M = \{I, C, B\}$, $N = \{c_1, c_2, b_1, b_2, \text{if}, \text{then}\}$, $R$ are the production rules (e.g., $C \to c_1$), and $I$ is the initial symbol. This language accepts programs such as if $b_1$ then $c_1$.

Given a domain-specific language $D$, our task is to find a program $p \in [\![D]\!]$ encoding a policy that maximizes the expected sum of discounted rewards for a given MDP. As assumed in previous work (Verma et al., 2018), depending on the language used, policies could be human-interpretable. We are interested in measuring, for a given policy $\pi$, the extent to which $\pi$ is human-interpretable.

$$I \to \text{if}(B) \text{ then } C$$
$$C \to c_1 \mid c_2$$
$$B \to b_1 \mid b_2$$

Figure 1: Grammar defining a DSL.

## 3  LINT Score

In this section, we present the LINT score, an intepretability metric for programmatic policies. Figure 2 shows how to compute the score. An LLM, the **explainer**, receives a programmatic policy $\pi$, a set of constraints $C$, and a description of the domain-specific language (DSL) in which $\pi$ was written. Then, it produces a natural language description of the behavior of $\pi$. Another LLM, the **reconstructor**, attempts to reconstruct the policy from the natural description the explainer generated for $\pi$. The LINT score of $\pi$ is then a behavior similarity metric $B$ of the original policy $\pi$ and the reconstructed policy $\pi'$. For example, $B$ can measure the action agreement of $\pi$ and $\pi'$ on a set of predefined states. We describe two domain-independent metrics $B$ in Section 4. If $\pi$ and $\pi'$ have similar behavior in terms of $B$, then $\pi$ is deemed interpretable by the LINT score.

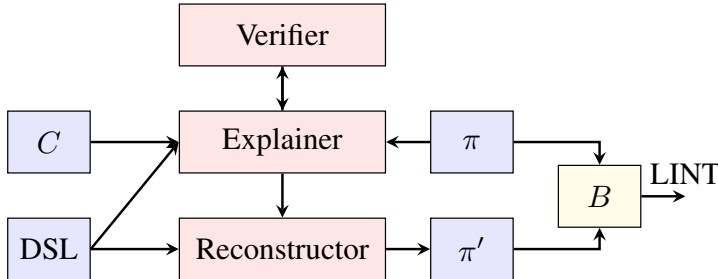

Figure 2: Overview of LINT. The Explainer receives a programmatic policy $\pi$, a set of constraints $C$, and a description of the domain-specific language in which $\pi$ was written; it produces a natural language explanation of the behavior $\pi$, while satisfying the constraints $C$. The constraints state that the explanation must be in natural language and about the policy behavior. The constraints also ensure that the explanation does not include step-by-step instructions on how to write $\pi$. The constraints are checked by the Verifier. The explanation is provided as input, with the description of the domain-specific language, to the Reconstructor, which attempts to reconstruct $\pi$ from the explanation, thus producing $\pi'$. $B$ is a behavior metric that scores the similarity between $\pi$ and $\pi'$.

The underlying assumption of the LINT score is that a policy is interpretable if one can reconstruct the policy from a natural language description of the policy behavior, and this description can only be produced by inspecting the program that encodes the policy. For example, it is unlikely that one can describe the behavior of an agent by simply inspecting the weights of a deep neural network. Even if this was possible, it is unlikely that one could directly set the weights of a new neural network from a natural language description of the behavior of the agent. Thus, programmatic policies that are neural networks are expected to have a small LINT score, indicating that they are not interpretable.

However, it is possible to design non-behavioral natural language descriptions of $\pi$ that could allow the reconstructor to perfectly reconstruct the original policy. For example, the explainer could describe the architecture of the neural network encoded in $\pi$ as well as the set of weights used in each layer of the network. The reconstructor could then simply reimplement this neural network from its natural language description. The set of constraints $C$ specify that the explainer is required to gener-

ate a behavioral description of the policy, and it is not allowed to provide step-by-step programming instructions that would allow the reconstructor re-generate $\pi$, even for non-interpretable policies $\pi$.

We use the following constraints, with their complete description in Appendices J and I.

1. The explanation of the program must be in natural language.
2. Do not use programming language jargon in your explanation.
3. Do not give line-by-line programming instructions of how to write the program.

The final component in Figure 2 is the **verifier**, which is another LLM whose purpose is to check if the constraints $C$ are met. The verifier checks if the explanation produces a non-behavioral natural language description of $\pi$, such as step-by-step programming instructions. If the verifier answers 'no' to satisfying the constraints, then we sample another explanation from the explainer, until they are satisfied; only then the LINT score is computed. Due to the stochastic nature of the LLMs, it is possible that the explainer fails to satisfy the constraints and that the verifier incorrectly flags the constraints as satisfied. We can increase the confidence that the constraints are satisfied by invoking the verifier multiple times, and use a voting mechanism to approve the explanation. In our experiments, we invoked the verifier only once, and all approved explanations satisfied the constraints.

Instead of computing the LINT score only once, in practice we computed it $k$ times and report the average score. This multi-evaluation approach is to account for the stochasticity of the LLMs involved in the process. The value of $k$ should be large enough to account for the variance of the LLMs and small enough to prevent them from reconstructing the original program by chance. Since the program spaces of practical interest are very large, it is unlikely that the reconstructor will generate programs similar to $\pi$ if $\pi$'s behavior cannot be accurately described in natural language.

## 4 EMPIRICAL METHODOLOGY

The objective of our evaluation is to evaluate our hypothesis that the LINT scores correlate with the interpretability of a set of programmatic policies. Our evaluation is divided into two parts. In the first part, we rely on methods from the static obfuscation literature (Collberg & Nagra, 2009) to generate programs with different levels of interpretability. Static obfuscation algorithms transform a program, before it starts running, into less interpretable programs, with the goal of making it harder for adversarial agents to gain knowledge of the program by reading its implementation. In the second part, we perform a user study with people with different levels of computer programming proficiency. In our study, people are presented different programs and asked to score the interpretability of the programs and also to answer a multiple-choice question that verifies their understanding of the program.

**Domains.** We use MicroRTS, a real-time strategy game (Ontañón, 2017), and Karel the Robot, a platform originally designed to teach computer programming (Pattis, 1994). We chose these domains because they are challenging and programmatic policies have been shown to perform well in them (Trivedi et al., 2021). They are also diverse in that they use different domain-specific languages and that MicroRTS is multi-agent, while Karel is single-agent. For MicroRTS, we consider policies synthesized with the self-play algorithm 2L (Moraes et al., 2023) for the map BaseWorkers-16×16A. For Karel, we consider policies synthesized with stochastic hill climbing (Carvalho et al., 2024) for the problems StairClimber, Maze, FourCorners, Seeder, and TopOff (Trivedi et al., 2021).

**Domain-Specific Languages.** Table 1 shows examples of programmatic policies, with one from MicroRTS on the left and one from Karel the Robot on the right. In MicroRTS, the program is invoked in each time step to define the action the agent will perform; in Karel, the program is invoked only once and it determines the action of the agent for all time steps. For MicroRTS, the language allows for loops that iterate over all units the player controls. It also offers several domain-specific instructions that check for conditions such as `canHarvest`, in line 3, which returns true if the unit can collect resources. The language also offers instructions that assign actions to units, such as `harvest`, in line 4, which will send the unit to collect resources. For Karel the Robot, the language has perception instructions such as `markersPresent`, in line 5, which checks whether the grid cell the agent occupies has a marker. The language also has action instructions such as `putMarker` and `turnRight`, in lines 2 and 6, respectively. Appendix B describes the domain-specific languages of each domain.

| MicroRTS | Karel the Robot |
|---|---|

```
1  for unit u
2      u.train(Worker, Right, 5)
3      if u.canHarvest()
4          u.harvest(2)
5      u.build(Barracks, EnemyDir, 8)
```

```
1  while noMarkersPresent
2      putMarker
3      move
4      move
5      while markersPresent
6          turnRight
7          move
```

Table 1: Examples of programmatic policies for MicroRTS (left) and Karel the Robot (right).

**Metrics $B$.** We use two behavior metrics: *action* and *return*. The action metric assumes discrete action spaces, and the return metric assumes a reward function. For the action metric, we compute the fraction in which the actions chosen by the original program $\pi$ match the actions chosen by the reconstructed program $\pi'$ for states in a set $S_\pi$: $|S_\pi|^{-1} \times \sum_{s \in S_\pi} \mathbf{1}[\pi(s) = \pi'(s)]$. If the reconstructed program is equivalent to the original, then the action metric is $1.0$. The return metric averages the absolute difference between the reward obtained by rolling out $\pi$ and $\pi'$ from a set of initial states $S_I$: $|S_I|^{-1} \times \sum_{s \in S_I} |R_{\pi,s} - R_{\pi',s}|$. Here, $R_{\pi,s}$ is the reward obtained by rolling out $\pi$ from $s$. Since MicroRTS and Karel are deterministic problems, $R_{\pi,s}$ is computed with one roll-out. If the reconstructed program is equivalent to the original, the return metric is $0.0$. In Sections 4.1 and 4.2 we explain how the sets $S_I$ and $S_\pi$ are obtained for the two parts of our experiments.

**Baselines.** Recent empirical research has highlighted the lack of an effective metric to measure code interpretability (Scalabrino et al., 2021). Consequently, no established baseline is specifically optimized for this task. To address this, we developed two baselines. The first baseline involves using an LLM to assign an interpretability score from 0 to 1 to the set of evaluated programs given as input to the model (LLM Baseline). We use with this LLM baseline prompts similar to those used to compute the LINT score (Appendix K). Also, similarly to LINT, we deal with the stochasticity of the LLM by running the baseline $k$ times; the interpretability score of the programs provided as input is the average across the $k$ calls to the LLM. We also consider as baseline a set of structural metrics, such as the number of lines of code, loops, and conditionals, to approximate interpretability. Appendix E shows that none of these metrics can reliably score the interpretability of programs.

**LLM Model** All experiments used GPT-4 (OpenAI et al., 2023) with a temperature of 0.7 and $k = 5$ for both LINT and the LLM baseline. Our prompts are given in Appendices J and I.

### 4.1 PART 1: EVALUATING OBFUSCATED PROGRAMMATIC POLICIES

In the first part of our evaluation, we make the reasonable assumption that obfuscated programs are less interpretable than non-obfuscated ones. Our hypothesis is that LINT scores will correlate (negatively for the action metric and positively for the return metric) with the level of obfuscation of a program. We use the obfuscation technique of adding useless code snippets to the programs (Cohen, 1993). We consider two levels of obfuscation: level 1, where we add code snippets with a few lines of code, and level 2, where we add longer code snippets that do not change the agent behavior.

Table 2 shows examples of the level-1 snippets we use; all snippets used in our experiments are shown in Appendix H. The left snippet in Table 2 does not change the behavior of any MicroRTS policy because the only unit that can harvest resources is a builder, so line 6 does not change the behavior of the policy. The right snippet is used in the StairClimber problem. This code does not change the agent's behavior because markersPresent only returns true in the last state of the problem, when the agent encounters a marker and its interactions with the environment finish.

For MicroRTS, we randomly sample 20 policies from the policies generated in the 2L self-play process; these are the policies $\pi$ for which we compute the LINT score. We sample 10 additional policies, which are different from the initial 20, from the same pool of 2L policies to serve as "opponent policies" $\pi_o$. The set $S_I$ is given by pairs $(s_0, \pi_o)$, where $s_0$ is the initial state of the map and $\pi_o$ is an opponent policy. We have one such pair in $S_I$ for each of the 10 opponents. The states

| MicroRTS | Karel the Robot |
|---|---|

```
1  if u.canHarvest():
2      for unit u
3          if u.isBuilder():
4              pass
5          else:
6              u.harvest(50)
```

```
1  while markersPresent
2      pickMarker
3      turnRight
4      turnRight
5      turnRight
```

Table 2: Code snippets that do not change the agent behavior for MicroRTS (left) and Karel (right).

encountered along the trajectory obtained by playing each policy $\pi$ in the set of 20 policies against each $\pi_o$ in the set of 10 policies form the sets $S_\pi$, one set for each $\pi$. For Karel, we synthesized one policy $\pi$ for each problem. These are the policies for which we evaluate LINT score. The sets $S_I$, one for each of the 5 problems we consider, each have a single initial state. The states encountered along the trajectories obtained by rolling out each $\pi$ from the start state form the sets $S_\pi$.

## 4.2 PART 2: USER STUDY

We enlisted 93 users in our study. We advertised our study in the mailing lists of two Computing Science departments from two different countries. Members of one department were invited to evaluate the interpretability of the MicroRTS policies, and members of the other department were invited to evaluate the Karel policies. For MicroRTS, we had a total of 60 participants, of which 50 were male and 10 were female. The average age was 23.66 years with a standard deviation of 5.87. On a scale of 1 to 4, where 1 means a beginner in computer programming and 4 means an expert, the average was 1.78 with a standard deviation of 0.86. For Karel, we had 33 participants, of which 22 were male and 11 were female. The average age was 23.37 years with a standard deviation of 3.79. The average response for programming experience was 2.68 with a standard deviation of 0.82.

For MicroRTS, we used two policies in the study. $\pi_1$ and $\pi_2$, randomly selected from the 20 used in the obfuscation experiment. In addition to these two policies, we also use their obfuscated versions, at level 2. The sets $S_I$ and $S_\pi$ are the corresponding sets for $\pi_1$ and $\pi_2$ described in Section 4.1. For Karel, we used the same set of policies $\pi$ and sets $S_I$ and $S_\pi$ used in the obfuscation experiments.

Each participant in our study had to agree to a consent form, which explained the risks involved in the study and that our study was approved by our institution's ethics board. They then answered the demographic questionnaire and were asked to read a tutorial on the problem domain (MicroRTS or Karel) and the domain-specific language of each domain. The participants then answered review questions about the problem domain and language, before going through the questions that evaluated the interpretability of the programs $\pi$. For MicroRTS, since the domain-specific language is different from most commonly used programming languages, we presented an example of a program and explained its behavior. This extra step was meant to ensure that participants understood the language before trying to answer the study questions.

We asked the users two questions for each policy $\pi$. The first question asked users to rate on a Likert scale of 1 to 5 how interpretable the program encoding $\pi$ was, where 1 means the least interpretable and 5 the most interpretable. Second, we presented the participants with a multiple-choice question in which each option of the question showed a video of an agent behavior (a match of MicroRTS or Karel interacting with the environment). The user had to then choose the video that best described the behavior of $\pi$. Although the interpretability question always came before the video question, the pair of questions for each policy $\pi$ were randomly shuffled to account for learning effects.

We used these two types of question because they may complement each other. Asking what the user perceives as interpretable has the advantage of being a direct question to what we want to measure. The disadvantage is that there could be a mismatch between what the user perceives as interpretable and what is actually interpretable to them. The video questions have the strength of evaluating what the user understood about the program, which contrasts with the user perception of the first question. The disadvantage of this type of question is that it is challenging to design multiple-choice questions

| Domain | Metric | Original Program | Obfuscation Level | |
|---|---|---|---|---|
| | | | Level 1 | Level 2 |
| **MicroRTS** | Action ↑ | 0.721 ±0.015 | 0.646 ±0.017 | 0.578 ±0.015 |
| | Return ↓ | 0.880 ±0.019 | 0.951 ±0.021 | 1.031 ±0.023 |
| **Karel** | Action ↑ | 0.327 ±0.072 | 0.182 ±0.045 | 0.088 ±0.013 |
| | Return ↓ | 0.750 ±0.092 | 1.026 ±0.072 | 1.071 ±0.069 |

Table 3: Average values of the action and return metrics for the original policy and for the two levels of obfuscated policies in MicroRTS and Karel domains; the cells also show the 95% confidence interval. Higher values of the action and lower values of the return metric mean more similar policies.

| Difficulty | Policy | LINT | I-Score | V-Score | LLM Baseline |
|---|---|---|---|---|---|
| **More Interpretable** | StairClimber | $0.932 \pm 0.090$ | $4.250 \pm 0.370^a$ | $0.750 \pm 0.128^a$ | $0.820 \pm 0.030$ |
| | Maze | $0.626 \pm 0.150$ | $4.031 \pm 0.327^a$ | $0.687 \pm 0.137^{ab}$ | $0.580 \pm 0.030$ |
| | **Average** | $0.779 \pm 0.087$ | $4.140 \pm 0.247$ | $0.718 \pm 0.094$ | $0.700 \pm 0.030$ |
| **Less Interpretable** | FourCorners | $0.012 \pm 0.008$ | $2.937 \pm 0.363^b$ | $0.562 \pm 0.147^{bd}$ | $0.660 \pm 0.020$ |
| | Seeder | $0.016 \pm 0.015$ | $2.531 \pm 0.331^c$ | $0.656 \pm 0.140^{ad}$ | $0.730 \pm 0.020$ |
| | TopOff | $0.046 \pm 0.006$ | $3.093 \pm 0.350^{bc}$ | $0.312 \pm 0.137^c$ | $0.720 \pm 0.020$ |
| | **Average** | $0.024 \pm 0.006$ | $2.853 \pm 0.201$ | $0.510 \pm 0.082$ | $0.703 \pm 0.020$ |

Table 4: User study results for Karel the Robot. The table presents the LINT scores, the average Likert scores for the interpretability question (I-Score, which range from 1 to 5), and the average number of times participants answered correctly the video-based multiple-choice question (V-Score, which range from 0 to 1). The policies categorized into more and less interpretable sets. The table also presents the 95% confidence interval, which was computed with 10 seeds for the LINT scores. The letter superscript for the I-Score and V-Score columns show statistical difference between the scores of the policies: rows with different letters indicate statistically different scores with $p < 0.05$. For more clarification on the superscripts and the significance tests, check Appendix L, Section L.1

with relevant distractors (Liu et al., 2017), that is, options that cannot be easily ruled out as wrong. In this way, users might choose the correct answer without being able to fully interpret the program.

## 5 EMPIRICAL RESULTS

In this section, we present the results for parts 1 (Section 5.1) and 2 (Section 5.2) of our evaluation.

### 5.1 RESULTS: OBFUSCATED PROGRAMMATIC POLICIES

Table 3 presents the results of part 1 of our evaluation. For the action metric, higher values indicate reconstructed policies that are more similar to the original policy. For the return metric, smaller values indicate more similar policies. The arrows in the table (↑ and ↓) indicate the relationship between the metrics. For MicroRTS, the average values shown in the table are computed over 200 executions of LINT: 10 for each of the 20 policies $\pi$ used in this experiment. For Karel, they are computed over 25 executions of LINT: 5 for each of the 5 policies. The 95% confidence intervals show that all differences are significant, with the exception of the return metric for Level 1 and Level 2 in Karel. The value of the action metric drops as we move from the original policies to the obfuscated ones. We also observe an increase in terms of the return metric, which also suggests that the LINT score can capture the decrease in interpretability as we increase the level of obfuscation.

| Policy Type | Policy | LINT | I-Score | V-Score | LLM Baseline |
|---|---|---|---|---|---|
| Original | $\pi_1$ | $0.754 \pm 0.052$ | $3.904 \pm 0.200^a$ | $0.555 \pm 0.104^a$ | $0.390 \pm 0.040$ |
| | $\pi_2$ | $0.835 \pm 0.041$ | $3.904 \pm 0.209^{ac}$ | $0.587 \pm 0.104^a$ | $0.680 \pm 0.030$ |
| Obfuscated | $\pi_1$ | $0.712 \pm 0.071$ | $3.396 \pm 0.217^{bc}$ | $0.492 \pm 0.103^a$ | $0.640 \pm 0.030$ |
| | $\pi_2$ | $0.695 \pm 0.050$ | $3.190 \pm 0.244^b$ | $0.396 \pm 0.102^a$ | $0.750 \pm 0.040$ |

Table 5: User study results for MicroRTS. The table presents the LINT scores, the average Likert scores for the interpretability question (I-Score, which range from 1 to 5), and the average number of times participants answered correctly the video-based multiple-choice question (V-Score, which range from 0 to 1). The policies are categorized into the original and obfuscated policies. The table also presents the 95% confidence interval, which was computed with 10 seeds for the LINT scores. The letter superscript for the I-Score and V-Score columns show statistical difference between the scores of the policies: rows with different letters indicate statistically different scores with $p < 0.05$. For more clarification on the superscripts and the significance tests, check Appendix L, Section L.2

## 5.2 Results: User Study

### 5.2.1 Karel the Robot

Table 4 shows the average results of the user study for Karel the Robot. Each row shows the average LINT score across 10 independent runs of the system, the average Likert value for the interpretability question (I-Score), and the average number of times a participant answered the video-based question correctly (V-Score). We split the 5 problems into two disjoint sets, the ones that are more interpretable (StairClimber and Maze) and the ones that are less interpretable (FourCorners, Seeder, and TopOff), so we can present the average LINT, I, and V scores for the programs in each group. The superscript letters indicate the results of a Friedman test, followed by pairwise Wilcoxon tests when the Friedman test is significant at $p < 0.05$. If two policies have statistically different scores in the Wilcoxon test ($p < 0.05$), they are assigned different letters. For instance, the V-Score of Maze is not statistically different from that of StairClimber, but it is significantly different from TopOff.

We also performed a Spearman correlation test between the LINT scores and the I-Scores and a point-biserial correlation test between LINT scores and the V-scores. The correlation between LINT and the I-Scores is higher ($0.447$) than between LINT and the V-Scores ($0.204$); both correlations are statistically significant with $p < 0.05$. The numbers in Table 4 offer an explanation for the correlation results. The LINT score closely followed the I-Score in the sense that higher LINT scores resulted in higher I-Scores, except for FourCorners and Seeder. The correlation is weaker with respect to the V-Score because we observe a few discrepancies in the results. For example, the LINT score considers the policy for Seeder to be less interpretable than the policy for Maze. However, Seeder and Maze have similar V-Scores. Based on feedback from a few participants, who mentioned being able to easily eliminate certain options in the multiple-choice question for Seeder, some of the discrepancies between the I-Scores and V-Scores may stem from participants confidently ruling out certain choices in the multiple-choice question. The elimination of choices for some of the policies increases the average V-Score of them comparatively with the policies for which it is more difficult to eliminate choices, thus causing the I-Scores and V-Scores discrepancies.

The average score and the 95% confidence interval of the LLM Baseline was computed over 10 independent runs of the system. A Spearman correlation test between the LLM baseline scores and the I-Scores was close to zero ($0.02$). Similarly, a point-biserial correlation between the LLM Baseline scores and the V-Scores was $-0.03$. These results contrast with the correlation noted in the LINT scores and highlight the importance of the "natural language bottleneck" LINT uses.

The I-Score and V-Score results of Table 4 also support our hypothesis that programmatic policies that were previously considered interpretable can significantly differ in their degree of interpretability. LINT can capture this difference when there is a large interpretability gap between policies. That is, if we observe the policies with the largest interpretability gap (e.g., StairClimber and FourCorners or StairClimber and TopOff), LINT correlates well both in terms of their I-Score and V-Score. This is also evidenced by the average results of the more and less interpretable groups. Mann-Whitney tests point to significant differences ($p < 0.05$) between the average I-Score and V-Score values.

### 5.2.2 MicroRTS

Table 5 shows the results of the MicroRTS user study. The correlation between LINT scores and I-Scores is $0.278$ (Spearman test) and between LINT scores and V-Scores is $0.126$ (point-biserial test); both tests are statistically significant with $p < 0.05$. The correlation results are weaker for MicroRTS because the interpretability gap between the policies is smaller in this domain, as can be observed by the smaller variance across I-Score and V-Score values. Despite the smaller gap, LINT could detect the decrease in interpretability for both $\pi_1$ and $\pi_2$ as we obfuscate the policies. This decrease in interpretability is also noted in both I-Score and V-Score. The differences between original and obfuscated are statistically different for I-Score, but not for the V-Score.

In general, the results of our user study suggest that LINT can distinguish the interpretability of programs with a sufficiently large intepretability gap. Examples of such distinctions include Stair-Climber and TopOff in Table 4 and the original programs $\pi_1$ and $\pi_2$ and their obfuscated counterparts in Table 5. However, LINT cannot distinguish the difference between FourCorners and Seeder, despite it being statistically significant in terms of I-Scores. It is reasonable that the LINT score cannot flag this distinction, as the difference in terms of I-Score between the two policies is small and possibly negligible in any practical application of interpretable policies.

Similarly to Karel, the average score and the 95% confidence interval of the LLM Baseline was computed with 10 independent runs of the baseline. A Spearman correlation test resulted in $-0.16$, and a point-biserial correlation test resulted in $-0.07$. Similarly to Karel, these results highlight the need of LINT's natural language bottleneck to approximate the interpretability of the programs.

Finally, as a side note, the results shown in Table 5 also support the assumption that we made in the first part of our experiments that obfuscated programs are less interpretable. In particular, there is a significant drop in I-Score as we move from the original to the obfuscated versions of the policies.

We present in Appendix C representative examples of the reconstruction process of LINT.

## 6 LIMITATIONS OF THE LINT SCORE

When assessing the interpretability of programs, we have to assume a level of knowledge of the person interpreting them and a bounded amount of thinking time. We will refer to the person who interprets the program as *the user*. LINT assumes the knowledge of an LLM, which may not reflect reality due to a mismatch of knowledge between the LLM and the user. As a result, a policy that is considered interpretable according to its LINT score might not necessarily be interpretable to the users. If the program requires knowledge that the LLM does not possess (e.g., the LLM may not be able to generate accurate natural language descriptions of linear transformations of the state), LINT can produce false negatives. Similar arguments can be made with respect to the pondering time of the user. A cursory read of a program might lead the user to a shallow understanding of what the policy does; a deeper analysis might lead to a deeper understanding of the policy behavior. The selection of the LLM used in the computation of the LINT score implicitly assumes a level of knowledge and pondering time that may not reflect the end users of the application domain.

Similarly to the BLEU score (Papineni et al., 2002), LINT should not be used as an objective function. At least in its current form, using LINT as such could cause the system to disregard $C$, and the explainer could generate non-interpretable explanations that the reconstructor can use to fully reconstruct the program. Future work may investigate how LINT can be used as an objective function while ensuring that the explainer generates human-interpretable descriptions of the programs.

## 7 RELATED WORK

In contrast to the literature on programmatic policies, it is common to find evaluations of the interpretability of models in the context of supervised learning (Ribeiro et al., 2016; Lundberg & Lee, 2017; Fong & Vedaldi, 2017). In addition to LINT, another contribution of our work is that we are the first to provide a systematic evaluation of the interpretability of programmatic policies.

Previous work in programmatic policies, such as NDPS (Verma et al., 2018), Propel (Verma et al., 2019), and $\pi$-PRL (Qiu & Zhu, 2022), describe systems that synthesize programmatic policies in the

space of oblique decision trees (Murthy et al., 1994). Such trees represent programs with if-then-else structures with linear transformations of the inputs. Although we did not evaluate oblique decision trees in our experiments, both LINT and participants with general programming experience would likely find them non-interpretable due to the complexity of chaining multiple linear functions. Since oblique trees are equivalent to ReLU networks (Lee & Jaakkola, 2020; Orfanos & Lelis, 2023), interpreting these trees would suggest that ReLU networks are also interpretable. However, expert users might be able to interpret small and sparse oblique trees, as seen in previous work. Extending LINT to model expert interpretation is a promising direction for future research.

Measurements of code understandability (Buse & Weimer, 2010; Posnett et al., 2011; Daka et al., 2015; Scalabrino et al., 2016; Oliveira et al., 2020) from the Software Engineering literature attempt to solve a similar problem from the interpretability of programmatic policies. Code understandability considers scenarios where people write computer code that is meant to be understandable by other people, but it may not be because the person reading the code is not familiar with the API being used or the API documentation is lacking (Scalabrino et al., 2021). This contrasts with our problem, where the programs are written by other programs with the goal of maximizing the agent's return. Despite this difference, future work might benefit from more tightly connecting these areas.

Recent empirical work showed that there is currently no effective metric to measure code understandability (Scalabrino et al., 2021). Although there is no similar study in the context of interpretable policies, the results of Scalabrino et al. (2021) suggest that simple metrics such as counting structural features (e.g., number of lines) also will not perform well in estimating policy interpretability. Appendix D shows examples of obfuscated programs that solve classical programming problems in which structural metrics, such as the number of lines, would fail to estimate program interpretability, while LINT is still able to flag obfuscated programs as non-interpretable. The results of Scalabrino et al. (2021) contrast with our results, since our results suggest that LINT can be used as a reliable metric to assess the interpretability of programmatic policies. Our encouraging results suggest that future research could investigate the use of LLMs to measure code understandability.

## 8    CONCLUSIONS

Programmatic policies are often synthesized with the expectation of interpretability. However, previous work has only presented anecdotal evidence of policy interpretability. The lack of a systematic evaluation of the interpretability of such policies through user studies is likely due to the high cost of these evaluations. In this paper, we introduced an inexpensive metric based on LLMs to assess the interpretability of programmatic policies. The LINT score of a programmatic policy is obtained by running the policy through a "natural language bottleneck". Our assumption is that a policy is interpretable if it can be reconstructed from a natural language description of its behavior. The LINT score is computed by comparing the behavioral similarity between the original policy and the LLM-reconstructed policy. This methodology allows LINT to assign different degrees of interpretability depending on the similarity of the behaviors of the reconstructed and original policies.

Our empirical evaluation of LINT relied on the literature on program obfuscation, where we assumed that obfuscated programs are less interpretable than non-obfuscated ones. We also conducted two user studies in which people with a programming background evaluated the interpretability of programs generated by current systems for MicroRTS and Karel. The results of our experiments supported our hypothesis that the LINT score can effectively estimate the interpretability of policies. Specifically, the LINT score decreased as the level of obfuscation in the policies increased. Furthermore, the results of the user studies showed that LINT scores correlate with how users rated the interpretability of the evaluated policies. LINT faced difficulties distinguishing between policies that were similarly interpretable. This is reasonable, as even human evaluators might find it challenging to differentiate between nearly equally interpretable policies. Our empirical results suggest that LINT could serve as a valuable empirical tool for advancing research in policy interpretability.

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

# A PROBLEM DOMAINS

## A.1 KAREL

Karel the Robot is a grid world problem where the agent could perform different tasks. The domain was originally designed to teach computer programming (Pattis, 1994). Tasks include collecting all markers on a grid, or finding its way out of a maze. Figure 3 (right) shows a screenshot of the Maze problem. There is a marker on the bottom-left corner of the maze, and Karel is on the bottom-right corner, facing up. Next, we describe the 5 Karel problems used in our study.

1. **StairClimber** uses a $12 \times 12$ grid with stairs formed with walls. The agent starts on a random location on the stairs, and the goal is to collect a marker that is also randomly placed on the stairs. If the agent reaches the marker, the agent receives a reward of 1 and 0 otherwise. If the agent moves to any cell outside of the stairs, the program terminates with a reward of $-1$.

2. **Maze** uses a $8 \times 8$ grid, where a marker is placed on a randomly chosen cell. The agent starts on a random empty square of the grid and the goal is to find the marker. Once the agent finds the marker, it receives a reward of 1; the agent receives a reward of 0, otherwise.

3. **FourCorners** uses an empty $12 \times 12$ grid, where the agent starts in one of the cells of the bottom row. The goal is to place a marker in each corner of the grid. The reward is given the number of markers placed on a corner divided by four.

4. **Seeder** uses an empty $8 \times 8$ grid, where the agent is randomly placed in a cell. The goal is to place one marker in every cell of the grid. The reward is given by the number of cells with a marker divided by the number of cells.

5. **TopOff** uses an empty $12 \times 12$ grid where markers are randomly placed the bottom row of the grid. The agent starts at the bottom-left corner of the map and the goal is to place one marker on top of every existing marker in the grid. The reward is the number of markers that have been topped off divided by the total number of markers in the initial grid.

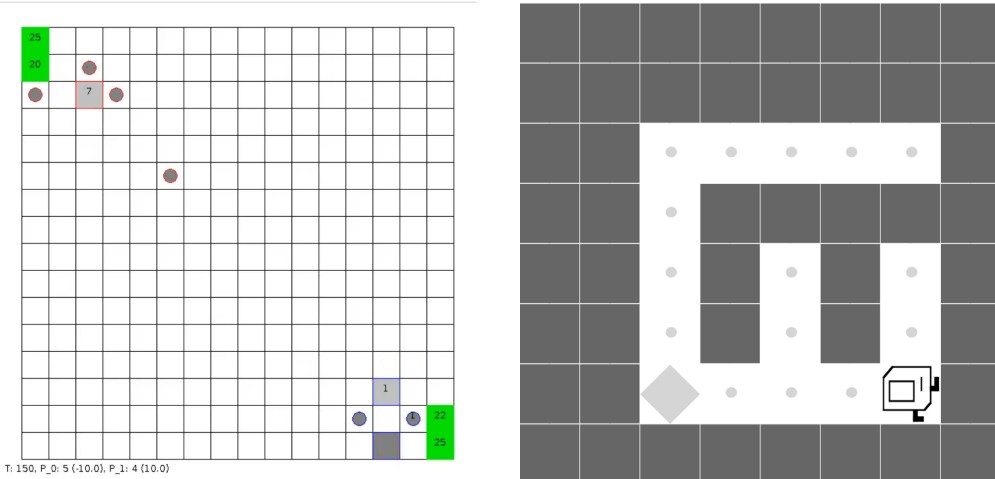

Figure 3: BaseWorkers-16×16A MicroRTS map (left) and the Maze problem for Karel (right).

## A.2 MICRORTS

MicroRTS is a minimalist two-player real-time strategy game used in research (Ontañón, 2020). In MicroRTS, each player controls a set of units in real time (each player is allowed 100 milliseconds

to decide on the next action). Units gather resources and build structures that are used to train more units that can fight more effectively against enemy units. MicroRTS has six types of unit: Worker, Ranged, Heavy, Light, Barracks, and Base. The first five types can move and battle the opponent, but only Workers can build structures and collect resources; the units differ in how much damage they can suffer before being removed from the game and how much damage they cause to opponent units. A Base can train Workers and store resources, while a Barracks can train the other units.

Figure 3 (left) shows a screenshot of a MicroRTS match on the BaseWorkers-16×16A map used in our experiments. One of the players is located on the top-left corner, while the other is located on the bottom-right corner. The light-gray squares represent Bases, while the dark-gray square represents a Barracks. The circles are Worker units, and the green squares are resources.

## B  DOMAIN-SPECIFIC LANGUAGES

Next, we present the domain-specific languages of Karel the Robot and MicroRTS. The following context-free grammar presents the language for Karel the Robot. This DSL is the same as that used in the previous work (Trivedi et al., 2021; Liu et al., 2023; Carvalho et al., 2024).

### B.1  KAREL THE ROBOT

$$
\begin{aligned}
\text{Program } \rho :=\ & \texttt{DEF run m(}\ s\ \texttt{m)} \\
\text{Statement } s :=\ & \texttt{WHILE c(}\ b\ \texttt{c) w(}\ s\ \texttt{w)}\ |\ \texttt{IF c(}\ b\ \texttt{c) i(}\ s\ \texttt{i)}\ | \\
& \texttt{IFELSE c(}\ b\ \texttt{c) i(}\ s\ \texttt{i) ELSE e(}\ s\ \texttt{e)}\ |\ \texttt{REPEAT R=}n\ \texttt{r(}\ s\ \texttt{r)}\ | \\
& s;s\ |\ a \\
\text{Condition } b :=\ & h\ |\ \texttt{not(}\ h\ \texttt{)} \\
\text{Number } n :=\ & 1\ |\ 2\ |\ 3\ |\ ...\ |\ \texttt{infinity} \\
\text{Perception } h :=\ & \texttt{frontIsClear}\ |\ \texttt{leftIsClear}\ |\ \texttt{rightIsClear}\ | \\
& \texttt{markersPresent}\ |\ \texttt{noMarkersPresent} \\
\text{Action } a :=\ & \texttt{move}\ |\ \texttt{turnLeft}\ |\ \texttt{turnRight}\ |\ \texttt{putMarker}\ |\ \texttt{pickMarker}
\end{aligned}
$$

```
1  while frontIsClear:
2      putMarker
3      move
```

Figure 4: An example of a program written in the domain-specific language for Karel.

The domain-specific language for Karel the Robot is meant to be intuitive and of easy use, as it was originally designed to teach computer programming to people. Consider the example shown in Figure 4. Given an MDP, the policy is invoked only once and will issue actions for each state that the agent encounters during its interaction with the environment. The policies written in this language have an internal state, which is the line of code of the program that will be executed next. The policy shown in Figure 4 places a marker and moves forward, until it reaches the wall or the end of the grid. The internal state of the policy will alternate between lines 2 and 3.

### B.2  MICRORTS

The following context-free grammar presents the language for MicroRTS, which was used in recent work (Moraes et al., 2023).

$$S \rightarrow SS \mid \texttt{for(Unit u) S} \mid \texttt{if(B) then S}$$
$$\mid \texttt{if(B) then S else S} \mid C \mid \lambda$$
$$B \rightarrow \texttt{hasNumberOfUnits}(T, N) \mid \texttt{opponentHasNumberOfUnits}(T, N)$$
$$\mid \texttt{hasLessNumberOfUnits}(T, N) \mid \texttt{haveQtdUnitsAttacking}(N)$$
$$\mid \texttt{hasUnitWithinDistanceFromOpponent}(N)$$
$$\mid \texttt{hasNumberOfWorkersHarvesting}(N)$$
$$\mid \texttt{is\_Type}(T) \mid \texttt{isBuilder()}$$
$$\mid \texttt{canAttack()} \mid \texttt{hasUnitThatKillsInOneAttack()}$$
$$\mid \texttt{opponentHasUnitThatKillsUnitInOneAttack()}$$
$$\mid \texttt{hasUnitInOpponentRange()}$$
$$\mid \texttt{opponentHasUnitInPlayerRange()}$$
$$\mid \texttt{canHarvest()}$$
$$C \rightarrow \texttt{build}(T, D, N) \mid \texttt{train}(T, D, N) \mid \texttt{moveToUnit}(T_p, O_p) \mid \texttt{attack}(O_p)$$
$$\mid \texttt{harvest}(N) \mid \texttt{attackIfInRange()} \mid \texttt{moveAway()}$$
$$T \rightarrow \texttt{Base} \mid \texttt{Barracks} \mid \texttt{Ranged} \mid \texttt{Heavy}$$
$$\mid \texttt{Light} \mid \texttt{Worker}$$
$$N \rightarrow 0 \mid 1 \mid 2 \mid 3 \mid 4 \mid 5 \mid 6 \mid 7 \mid 8 \mid 9$$
$$\mid 10 \mid 15 \mid 20 \mid 25 \mid 50 \mid 100$$
$$D \rightarrow \texttt{EnemyDir} \mid \texttt{Up} \mid \texttt{Down} \mid \texttt{Right} \mid \texttt{Left}$$
$$O_p \rightarrow \texttt{Strongest} \mid \texttt{Weakest} \mid \texttt{Closest} \mid \texttt{Farthest}$$
$$\mid \texttt{LessHealthy} \mid \texttt{MostHealthy} \mid \texttt{Random}$$
$$T_p \rightarrow \texttt{Ally} \mid \texttt{Enemy}$$

```
1  for(Unit u):
2      u.build(Barracks,Up,1)
3  for(Unit u):
4      u.attack(Closest)
```

```
1  for(Unit u):
2      u.train(Ranged, Right, 4)
3      u.harvest(9)
4      u.attack(Closest)
5      u.train(Worker, Up, 1)
```

Figure 5: Two examples of programs written in the domain-specific language for MicroRTS.

Consider the policy written in the MicroRTS domain-specific language in Figure 5 (left). The program is invoked in every iteration of the game to determine the action of each unit the player controls. In the example above, the outer for-loop iterates over all units the player controls. The instruction `u.build(Barracks, Up, 1)`, attempts to build a Barracks with unit `u`. Note that this is possible only if `u` is a Worker unit. If `u` is not a Worker unit, then the instruction will be ignored. Generally, any line of code that is not applicable to a unit will simply be skipped in this language.

The program also has an inner for loop, which will also go over all units the player controls. The instructions inside the inner for loop receive higher priority than instructions outside it. This is because, in this language, once an action is assigned to a unit, it cannot be changed. In this example, the inner for loop attempts to assign an action to all units that can attack. The language considers a fixed order of the type of units in which the for loop iterates over: Workers, Barracks, Base, Light, Ranged, and Heavy. If the player controls one Base, one Barracks, and four Workers, the Worker will be the first unit to be iterated over in the for loop, then the Barracks, and finally the Base.

Now, consider the example shown in Figure 5 (right). In this policy, if `u` is a Base unit, then only `u.train(Worker, Up, 1)` is applicable to it; all the other instructions are ignored. If `u` is a Worker unit, then the only commands applicable to it are `u.harvest(9)` and `u.attack(Closest)`. Once 9 Workers are assigned for harvesting, future calls to this function will ignore the harvest function for Workers; these Workers will be assigned the action given by `u.attack(Closest)`, meaning they will attack the closest enemy unit.

```
1  while noMarkersPresent          1  while not markersPresent
2      if rightIsClear             2      if rightIsClear
3          turnRight               3          turnRight
4      else                        4          move
5          while not frontIsClear  5      else
6              turnLeft            6          if frontIsClear
7      move                        7              move
                                   8          else
                                   9              turnLeft
```

Table 6: Original policy for the Maze problem (left) and a reconstructed policy (right).

## C    REPRESENTATIVE EXAMPLES OF THE RECONSTRUCTION PROCESS

### C.1    KAREL THE ROBOT

In this section, we present a representative example from the Karel domain of the natural language description generated by the explainer and the reconstructed policy. The original policy is shown in Table 6 (left). This is the synthesizer's solution to the Maze problem, where Karel the Robot navigates through a maze to find a marker. The natural language description that the explainer generated is given in Explanation 1. The explanation does not provide step-by-step programming instructions, but describes the behavior of the agent from the policy in Table 6 (left).

**Explanation 1.** *The given program instructs Karel the Robot to navigate its environment until it encounters a marker. Karel prioritizes turning right whenever the path to the right is clear. If the path to the right is blocked, Karel will begin turning left until it has a clear path in front. Once the front path is clear, Karel moves forward. At any point during this navigation, if Karel encounters a marker, it stops, completing its mission. Thus, the program's main goal is to navigate Karel through its environment by prioritizing right turns, then left turns, until a marker is found.*

The reconstructed policy is not syntactically identical to the original policy, as one would expect due to the LINT's natural language "bottleneck". However, the two policies are semantically identical. For example, instead of using `noMarkersPresent` as the Boolean expression of the while loop, the reconstructed policy uses `not markersPresent`, which is logically equivalent. If `rightIsClear`, the reconstructed program not only turns Karel to the right as in the original, but also applies the `move` command in the same block. This is equivalent to the `turnRight` instruction in line 3 of the original policy followed by the `move` instruction in line 7. Similarly, the outer else block of the reconstructed policy is equivalent to the inner while loop of the original policy.

The LINT score for the reconstruction discussed in this example indicates a perfectly interpretable program (action and return metrics of $1.0$ and $0.0$, respectively). We do not observe such scores for $k = 5$ because the system is unable to perfectly reconstruct the program in all $5$ runs. The LINT scores with $k = 5$ still point to an interpretable policy (action and return metrics of $0.626$ and $0.570$).

### C.2    MICRORTS

```
1  for(Unit u)                     1  for(Unit u):
2      for(Unit u)                 2      u.train(Worker,Up,2)
3          u.train(Worker,Up,2)    3      u.attack_if_in_range()
4      u.attack_if_in_range()      4      for(Unit u):
5      u.train(Heavy,EnemyDir,8)   5          u.train(Heavy,EnemyDir,8)
6  for(Unit u)                     6      for(Unit u):
7      u.train(Light,Left,100)     7          u.train(Light,Left,100)
8      u.build(Barracks,EnemyDir,1)8      u.build(Barracks,EnemyDir,1)
9      u.harvest(25)               9      u.harvest(25)
10     u.attack(Closest)          10      u.attack(Closest)
```

Figure 6: Original policy for the BaseWorkers-16×16A (left) and a reconstructed policy (right).

Next, we present a representative example from the MicroRTS domain of the natural language description generated by the explainer and the reconstructed policy. The original policy is shown in Figure 6 (left). The description that the explainer generated is given in Explanation 2. Similarly to the Karel example in Section C.1, the explanation does not provide step-by-step programming instructions, but describes the behavior of the agent from the policy in Figure **??** (left).

**Explanation 2.** *The script outlines a two-tiered strategy in the game of MicroRTS. It focuses on the tactical management of different types of units, ranging from Workers, to Heavy units, and finally Light units. The first part of the strategy involves a layered assignment of tasks. Every unit is asked to train two Worker units if possible. These Workers can be seen as the backbone of your army since they are the only units that can build structures, which in turn produce other units. At the same time, if a unit can't train a Worker, it is assigned to remain idle, thus serving as a defensive guard to your base. In addition, every unit is asked to train up to 8 Heavy units, that typically are stronger and can inflict more damage, acting as the frontline offensive unit. This phase essentially sets up your base with a balance of productive Workers and Heavy units.*

*In the second part of the strategy, each unit is tasked with producing as many Light units as possible, up to a whopping 100. Light units are generally quicker and can swiftly react to the changes in the battlefield. Additionally, every unit is tasked to build a Barracks near the enemy's direction, effectively establishing a forward base closer to the enemy territory. They are also asked to harvest resources, up to 25 units, ensuring a steady supply for creating more units and structures. Finally, units are directed to attack the closest enemy units, marking an aggressive stance towards the opponent.*

*The overall goal of this strategy is to ensure a robust base with a mixture of Worker and Heavy units, while also maintaining an aggressive stance with a large army of Light units and Barracks near the enemy territory. It aims at resource gathering for continued production of units and structures, and pushing the opponent back through relentless attack.*

The explanation describes the behavior of the policy in a level of detail that allows for a perfect reconstruction of the behavior of the original policy. Figure 6 (right) shows the reconstructed policy. The reconstructed policy is not syntactically identical to the original one, due to the information lost in the process of translating the policy into a natural language explanation and back to a policy.

A casual inspection of the policies might even suggest that the reconstructed policy does not behave as the original. This is because the original policy has an instruction for training Worker units inside a nested loop, thus giving it a higher priority. The reconstructed policy has the instruction for training Worker units instruction inside the main loop. This means that the reconstructed policy can use the player's resources to other functions (e.g., train Light units in line 7) and by the time $u$ is a Base in the outer loop, the player no longer has resources for training Worker units. However, a more careful inspection of the policy reveals that, in the first states of the game, where the player trains Worker units, none of the actions that use resources can be assigned to a unit: the player cannot train Heavy and Light units (lines 5 and 7, respectively) because the player does not have a Barracks yet; the player cannot build a Barracks (line 8) because it does not have enough resources to do so. Thus, similarly to the original policy, the reconstructed one prioritizes the training of Worker units.

## D  CLASSICAL PROGRAMMING PROBLEMS

Besides our primary domains, we also tested an additional domain in our study: "C Classical Programming Problems."

We consider 10 programs written in C for solving the following problems: computation of factorials, addition of two numbers, conversion of byte to binary, computation of all proper subsets of a set of arguments, of the value of $\pi$, of $\ln(n)$ for any $n$, of the smallest 100 prime numbers, of the square root of a number, sorting elements, and a program to play tic-tac-toe. The obfuscated versions of these programs were designed so that they would be as non-interpretable as possible, since all obfuscated programs we use are winning entries of the International Obfuscated C Code Contest IOCCC (1984). The obfuscated programs were constructed using different techniques, such as replacing sequences of instructions with equivalents that are less interpretable Cohen (1993). Figure 7 shows an example of the programs used in our experiment, where the first function is a non-obfuscated implementation for computing the proper subsets of a set of numbers, while the second is an obfus-

```
1   void subsets(char *av[], int c, int n, char *sbset[], int sz) {
2       if (c == n) {
3           if (sz < n) {
4               for (int i = 0; i < sz; i++)
5                   printf(sbset[i]);
6               printf("----------");
7           }
8           return;
9       }
10      sbsets(av, c+1, n, sbset, sz);
11      sbset[sz] = av[c];
12      sbsets(av, c+1, n, sbset, sz+1);
13  }
14
15  main(Q,O)char**O;{if(--Q){main(Q,O);O[Q][0]^=0X80;for(O[0][0]=0;O[++O
        [0][0]]!=0;)if(O[O[0][0]][0]>0)puts(O[O[0][0]]);puts("----------")
        ;main(Q,O);}}
```

Figure 7: Non-obfuscated code for computing proper subsets (lines 1–13); an obfuscated program for the same problem (line 15).

cated implementation to solve the same problem. The proper subsets of a set $I$ include all subsets except $I$.

The function $B$ we consider in this experiment measures the number of input values that the reconstructed program correctly maps to their corresponding output value. A $B$-value of $1.0$ indicates that the reconstructed program mapped all inputs to the correct output; a value of $0.0$ indicates that the reconstructed program failed on all inputs.

Here is the result of running C programming Problems through LINT: The system was able to reconstruct all non-obfuscated programs to solve the classical programming problems and naturally achieved the score of $1.0$. The system was unable to reconstruct any of the obfuscated programs, resulting in a score of $0.0$. As a representative example, Explanation 3 shows the natural language explanation of the Explainer for the non-obfuscated program shown in Figure 7.

**Explanation 3** (Proper Subset - Non-Obfuscated). *Imagine you have a list of items, and you want to figure out all the different ways you can select some of these items without taking all of them at once. When the program starts, it first checks if you've given it any items to work with. If not, it asks for some. Once it has the list of items, it begins to explore all the possible selections. It does this by starting with the first item and deciding whether to include it in the current selection or not. If it decides not to include the first item, it moves to the next item and makes the same decision. It continues doing this until it has considered every item in the list. After considering all the items, if it has left out at least one item from the current selection, it displays that selection for you to see. Then, it goes back and tries a different combination by including some items that it previously left out and leaving out some that it previously included. It continues this process until it has explored and shown you all possible selections of items from the list, always making sure it never shows you a selection that includes all the items at once. In the end, you're left with a display of all the different ways you can select some of the items without taking all of them.*

The explanation of the program is high-level and makes an analogy with a list of items, and the process of computing the proper subsets is described as a process of "selecting" and "displaying" the items, where all possible combinations of subsets are selected: *"It does this by starting with the first item and deciding whether to include it in the current selection or not"*. The Reconstructor can produce a correct implementation of a program that computes the proper subsets from this explanation. Explanation 4 shows the explanation of the obfuscated program in Figure 7.

**Explanation 4** (Proper Subset - Obfuscated). *Imagine you have a list of items. Initially, all items are marked as "normal." The program goes through the following steps:*

1. *It starts at the end of the list and works its way to the beginning.*

   - *For each item, the program does two actions:First, it keeps the item as "normal" and then displays all the items marked as "greater than normal."*

- *After displaying these items, it separates them with a line of dashes.*
- *Second, it switches the item from "normal" to "greater than normal" and repeats the display process.*

2. *After dealing with an item, the program moves to the next item closer to the beginning of the list and repeats step 2.*

*This process continues until the program has considered all items in the list.*

This explanation is well-structured and includes the steps that supposedly need to be performed. However, the description is not clear in some parts. For example, it is not clear what "greater than normal" means. The explanation also includes unimportant parts, such as describing that the items should be separated with a line of dashes.

## E  STRUCTURAL METRICS - BASELINE

The key motivation for proposing LINT comes from the lack of a single metric that accurately captures the interpretability of computer programs (Scalabrino et al., 2021). In this section, we present empirical evidence that simple structural metrics, such as line count, presence of loops, or conditionals, are insufficient as they often fail to reflect the interpretability of programs.

Table 7 presents a set of programs along with simple structural metrics that can, in principle, be used to estimate program interpretability. We consider the number of lines of code (Lines), number of loops (Loops), where the value of $d$ specifies how many nested loops the program has, and the number of conditionals (Conds), where the value of $d$ also specifies the number of nested conditionals.

The set of programs includes winning entries from the IOCCC (International Obfuscated C Code Contest) (IOCCC, 1984), which were designed to be as non-interpretable as possible and were judged by human experts (upper part of Table 7); we also include their non-obfuscated counterparts. The set also includes the programs used in our MicroRTS study (middle part), and the programs used in our Karel study (bottom part). We assume that the I-score of the contest-winning entries is 0, which is equivalent to assuming them to be as non-interpretable as possible. The non-obfuscated versions of the C programs receive an I-score of 1, indicating fully interpretable programs, as they all represent programs for solving simple programming tasks. The I-scores for the C programs thus contrast with what we present for MicroRTS and Karel, which were obtained through our user studies. We present the non-obfuscated C programs in Section E.1 and the obfuscated ones in Section E.2.

None of the simple structural metrics can capture the interpretability score, as given by the I-Score. For example, while having more lines negatively correlates with the I-Score for both MicroRTS and Karel programs, the number of lines of code positively correlates with the I-Score for the C programs. A similar story is observed for the number of conditionals with $d = 0$. The LINT score is the only metric that is able to capture the non-interpretability of the obfuscated C programs and the nuanced I-Score for the MicroRTS and Karel programs.

### E.1  NON-OBFUSCATED C PROGRAMS

**P1:** This program computes all proper subsets of the set of arguments passed to it.

```c
#include <stdio.h>

void generate_subsets(char *argv[], int current, int n, \
char *subset[], int subsetSize) {
    if (current == n) {
        if (subsetSize < n) {
            for (int i = 0; i < subsetSize; i++) {
                printf("%s ", subset[i]);
            }
            printf("\n");
        }
```

| | Program | Lines | Loops $(d = 0)$ | Loops $(d = 1)$ | Loops $(d = 2)$ | Conds $(d = 0)$ | Conds $(d = 1)$ | LINT Score | I-Score |
|---|---|---|---|---|---|---|---|---|---|
| **C Programs** | P1 | 18 | 1 | 0 | 0 | 2 | 1 | 1 | 1 |
| | P2 | 11 | 1 | 0 | 0 | 1 | 0 | 1 | 1 |
| | P3 | 16 | 1 | 1 | 0 | 2 | 0 | 1 | 1 |
| | P1-Obf | 9 | 1 | 0 | 0 | 1 | 1 | 0 | 0 |
| | P2-Obf | 6 | 0 | 0 | 0 | 0 | 0 | 0 | 0 |
| | P3-Obf | 4 | 0 | 0 | 0 | 0 | 0 | 0 | 0 |
| **MicroRTS** | $\pi_1$ | 11 | 1 | 3 | 0 | 0 | 0 | 0.754 | 3.904 |
| | $\pi_2$ | 11 | 2 | 2 | 0 | 0 | 0 | 0.835 | 3.904 |
| | $\pi_1'$ | 23 | 1 | 4 | 0 | 2 | 3 | 0.712 | 3.396 |
| | $\pi_2'$ | 23 | 2 | 3 | 0 | 2 | 3 | 0.695 | 3.190 |
| **Karel** | StairClimber | 5 | 0 | 1 | 0 | 0 | 0 | 0.932 | 4.250 |
| | Maze | 7 | 0 | 1 | 1 | 0 | 1 | 0.626 | 4.031 |
| | FourCorners | 12 | 0 | 0 | 2 | 1 | 1 | 0.012 | 2.937 |
| | Seeder | 12 | 0 | 0 | 2 | 1 | 1 | 0.016 | 2.531 |
| | TopOff | 14 | 0 | 1 | 2 | 0 | 1 | 0.046 | 3.093 |

Table 7: Comparison of structural metrics, LINT scores, and I-Scores on selected programs from C programming problems domain and our user study with programmatic policies. The value of $d$ indicates the level of nesting, with $d = 0$ representing single loops or conditionals, and $d = 1$ and $d = 2$ represent nested loops and conditionals.

```
        return;
    }

    generate_subsets(argv, current + 1, n, subset, subsetSize);
    subset[subsetSize] = argv[current];
    generate_subsets(argv, current + 1, n, subset, subsetSize + 1);
}

int main(int argc, char *argv[]) {
    if (argc <= 1) {
        printf("Please provide some elements to generate subsets.");
        return 1;
    }
    char *subset[argc-1]; // This will store the current subset
    printf("Proper subsets:\n");
    generate_subsets(argv+1, 0, argc-1, subset, 0);

    return 0;
}
```

**P2:** This program calculates the value of $\pi$.

```
#include <stdio.h>
#include <stdlib.h>

long insideCircleCount = 0, totalPoints = 0;

void computePi() {
    for (int i = 0; i < 10000; i++) {
        double x = (double) rand() / RAND_MAX;
        double y = (double) rand() / RAND_MAX;
```

```
            if (x * x + y * y <= 1) {
                insideCircleCount++;
            }

            totalPoints++;
        }
    }

    int main() {
        computePi();
        printf("%1.3f\n", 4. * insideCircleCount / totalPoints);
        return 0;
    }
```

**P3:** This program prints the input string sorted alphabetically.

```
#include <stdio.h>
#include <string.h>

void sortStringAlphabetically(char str[]) {
    int len = strlen(str);
    char temp;
    for (int i = 0; i < len - 1; i++) {
        for (int j = 0; j < len - i - 1; j++) {
            if (str[j] > str[j + 1]) {
                // Swap characters if they are out of order
                temp = str[j];
                str[j] = str[j + 1];
                str[j + 1] = temp;
            }
        }
    }
}

int main() {
    char inputString[100];

    printf("Enter a string: ");
    fgets(inputString, sizeof(inputString), stdin);

    // Remove the newline character at the end of the string
    if (inputString[strlen(inputString) - 1] == '\n') {
        inputString[strlen(inputString) - 1] = '\0';
    }

    sortStringAlphabetically(inputString);

    printf("Sorted string: %s\n", inputString);

    return 0;
}
```

## E.2 Obfuscated C Programs

**P1:** This program computes all proper subsets of the set of arguments passed to it.

```c
#include <stdio.h>
#include <stdlib.h>

main(Q,O)char**O;{
    if(--Q){
        main(Q,O);
        O[Q][0]^=0X80;
        for(O[0][0]=0;O[++O[0][0]]!=0;)
            if(O[O[0][0]][0]>0)puts(O[O[0][0]]);
        puts("----------");
        main(Q,O);
    }
}
```

**P2:** This program calculates the value of $\pi$.

```c
#include <stdio.h>
#include <stdlib.h>

#define \_ F-->00 || F-OO--;

long F=00,OO=00;

F\_OO()
{
               _-_-_-_
           _-_-_-_-_-_-_-_
         _-_-_-_-_-_-_-_-_-_
        _-_-_-_-_-_-_-_-_-_-_
      _-_-_-_-_-_-_-_-_-_-_-_
      _-_-_-_-_-_-_-_-_-_-_-_
     _-_-_-_-_-_-_-_-_-_-_-_-_
    _-_-_-_-_-_-_-_-_-_-_-_-_
   _-_-_-_-_-_-_-_-_-_-_-_-_-_
   _-_-_-_-_-_-_-_-_-_-_-_-_
   _-_-_-_-_-_-_-_-_-_-_-_-_-_
    _-_-_-_-_-_-_-_-_-_-_-_-_-_
     _-_-_-_-_-_-_-_-_-_-_-_-_
      _-_-_-_-_-_-_-_-_-_-_-_
        _-_-_-_-_-_-_-_-_-_-_
          _-_-_-_-_-_-_-_-_
             _-_-_-_-_-_-_
                _-_-_-_
}
main(){F_OO();printf("%1.3f\n", 4.*-F/OO/OO);}
```

**P3:** This program prints the input string sorted alphabetically.

```c
#include "stdio.h"
main(argc, argv)
int argc;
char **argv;
{
while (*argv != argv[1] && (*argv = argv[1]) && (argc = 0) ||
(*++argv && (**argv && ((++argc)[*argv] && (**argv <= argc[*argv]
```

```
||(**argv += argc[*argv] -= **argv = argc[*argv] - **argv)) &&
--argv || putchar(**argv) && ++*argv--) || putchar(10))));
}
```

## F  SET OF KAREL PROGRAMS IN THE USER STUDY

### F.1  STAIRCLIMBER

```
DEF run m(
    WHILE c( noMarkersPresent c) w(
        turnLeft
        move
        turnRight
        move
    w)
m)
```

### F.2  MAZE

```
DEF run m(
    WHILE c( noMarkersPresent c) w(
        IFELSE c( rightIsClear c) i(
            turnRight i)
        ELSE e(
            WHILE c( not c( frontIsClear c) c) w(
                turnLeft w)
            e)
        move w)
    m)
```

### F.3  FOURCORNERS

```
DEF run m(
    IF c( leftIsClear c) i(
        WHILE c( leftIsClear c) w(
            move
            IF c( rightIsClear c) i(
                turnLeft
                move
                turnLeft
                turnLeft
                move
                putMarker i)
            WHILE c( frontIsClear c) w(
                move w)
            w)
        i)
    m)
```

### F.4 SEEDER

```
DEF run m(
    turnLeft
    WHILE c( noMarkersPresent c) w(
        putMarker
        REPEAT R=10 r(
            move r)
        REPEAT R=5 r(
            WHILE c( markersPresent c) w(
                turnLeft
                move
                turnRight w)
            pickMarker r)
    w)
    WHILE c( frontIsClear c) w(
        turnLeft w)
    m)
```

### F.5 TOPOFF

```
DEF run m(
    move
    REPEAT R=10 r(
        IF c( markersPresent c) i(
            WHILE c( rightIsClear c) w(
                move
                move
                w)
            putMarker
            WHILE c( not c( leftIsClear c) c) w(
                turnLeft
                move
                turnLeft
                move
                turnRight
                w)
            i)
        move r)
    m)
```

## G   SET OF MICRORTS PROGRAMS IN THE USER STUDY

```
π₁ - Original

for(Unit u){
    for(Unit u){
        u.harvest(2)
        u.idle()
    }
    u.train(Light,Up,25)
    u.train(Worker,Right,4)
    u.attack(Closest)
    for(Unit u){
```

```
            u.train(Heavy,Up,3)
        }
        for(Unit u){
            u.build(Barracks,Up,1)
        }
    }
```

**π₂ - Original**

```
for(Unit u){
    u.train(Heavy,EnemyDir,6)
    u.train(Light,EnemyDir,4)
    u.build(Barracks,Down,3)
    u.idle()
    u.train(Worker,Left,3)
}
for(Unit u){
    for(Unit u){
        u.harvest(15)
    }
    for(Unit u){
        u.moveToUnit(Enemy,Weakest)
    }
}
```

**π₁ - Obfuscated**

```
for(Unit u){
 for (Unit u){
  if (u.is_Type(Worker)) then{
   u.train(Heavy, Enemydir,10)
   if (u.canAttack()) then {
    u.train(Ranged,Up,16)
   }
  }
 }
 for(Unit u){
  u.harvest(2)
  u.idle()
 }
 u.train(Light,Up,25)
 u.train(Worker,Right,4)
 u.attack(Closest)
 for(Unit u){
  u.train(Heavy,Up,3)
 }
 if (u.canHarvest()) then {
  for (Unit u){
   if (u.is_Type(Ranged)) then{
     u.harvest(50);
   }
   else{
    if (u.is_Type(Heavy)) then{
     u.train(Light,Up,2)
    }
```

```
      }
     }
   }
  for(Unit u){
   u.build(Barracks,Up,1)
  }
 }
```

$\pi_2$ - **Obfuscated**

```
for(Unit u){
 if (u.canHarvest()) then {
  for (Unit u){
   if (u.is_Type(Ranged)) then{
     u.harvest(5);
   }
   else{
    if (u.is_Type(Heavy)) then{
     u.train(Light,Up,2)
    }
   }
  }
 }
 u.train(Heavy,EnemyDir,6)
 u.train(Light,EnemyDir,4)
 u.build(Barracks,Down,3)
 u.idle()
 u.train(Worker,Left,3)
}
for (Unit u){
 if (u.is_Type(Worker)) then{
  u.train(Heavy, Enemydir,2)
  if (u.canAttack()) then {
   u.train(Ranged,Up,1)
  }
 }
}
for(Unit u){
 for(Unit u){
  u.harvest(15)
 }
 for(Unit u){
  u.moveToUnit(Enemy,Weakest)
 }
}
```

# H  USELESS CODE SNIPPETS ADDED FOR OBFUSCATION

## H.1  MICRORTS

We added 10 and 35 lines for levels 1 and 2 obfuscation respectively.

### H.1.1 LEVEL 1

```
if (u.canHarvest()) then {
    for (unit u){
        if (u.isBuilder()) then{
        }
        else{
            u.harvest(50);
        }
    }
}
for (Unit u){
    if (u.is_Type(Worker)) then{
        u.train(Heavy, Enemydir, 10);
        if (u.canAttack()) then {
            u.train(Ranged, Up, 16);
        }
    }
}
```

### H.1.2 LEVEL 2

```
if (u.canHarvest()) then {
    for (unit u){
        u.train(Heavy,Left, 9)
        if (u.isBuilder()) then{

        }
        else{
            u.harvest(50);
        }
    }
}
for (Unit u){
    if (u.is_Type(Worker)) then{
        u.train(Heavy, Enemydir,10);
        if (u.canAttack()) then {
            u.train(Ranged,Up,16)
            if (u.canHarvest()) then {
                u.train(Worker,Right, 9)
                if (u.isBuilder()) then{
                    if (u.is_Type(Barracks)) then{
                        u.harvest(10);
                        u.train(Workers, Right, 2);
                    }
                    else{
                        u.train(Light,Down,5)
                    }
                }

            }
        }
    }
    else{
        u.harvest(1);
    }
```

```
    for(Unit u){
        u.idle()
    }
}
for (Unit u){
    if(u.hasLessNumberOfUnits(Heavy, 2)){
        if (u.isBuilder()) then{
            u.train(Light, Farthest, 1);
        }
        else{
            u.harvest(3);
        }

    }
    if (u.hasNumberOfUnits(Base, 3)){
        if (u.is_Type(Barracks)) then{
            u.build(Base,EnemyDir,1);
            u.moveToUnit(Ally, MostHealthy);
        }
        else{
            u.train(Heavy, Random,10);
        }
    }
}
```

## H.2 KAREL

The number of useless lines added for level one is around 2-3 lines and for level 2 is around 5-6 lines. The **bolded** code below represents the useless code that is added.

### H.2.1 STAIRCLIMBER

**Level 1** :

```
DEF run m(
    WHILE c( noMarkersPresent c) w(
        turnLeft
        IF c( markersPresent c)
            i( pickMarker i)
        move
        turnRight
        move w)
    m)
```

**Level 2** :

```
DEF run m(
    WHILE c( noMarkersPresent c) w(
        turnLeft
        WHILE c( markersPresent c) w(
            pickMarker
            turnRight
            turnRight
            turnRight w)
        move
        turnRight
        move w)
    m)
```

### H.2.2  MAZE

**Level 1**  :

```
DEF run m(
    WHILE c( noMarkersPresent c) w(
        IFELSE c( rightIsClear c) i(
            WHILE c( not c( noMarkersPresent c) c) w(
                putMarker w)
            turnRight i)
        ELSE e(
            WHILE c( not c( frontIsClear c) c) w(
                turnLeft w)
            e)
        move w)
    m)
```

**Level 2**  :

```
DEF run m(
    WHILE c( noMarkersPresent c) w(
        IFELSE c( rightIsClear c) i(
            WHILE c( not c( noMarkersPresent c) c) w(
                putMarker w)
            turnRight i)
        ELSE e(
            WHILE c( not c( frontIsClear c) c) w(
                IF c( rightIsClear c) i(
                    turnRight i)
                turnLeft w)
            e)
        move w)
    m)
```

### H.2.3  FOURCORNERS

**Level 1**  :

```
DEF run m(
    IF c( leftIsClear c) i(
        WHILE c( leftIsClear c) w(
            move
            IF c( rightIsClear c) i(
                turnLeft
                move
                turnLeft
                turnLeft
                move
                putMarker i)
            WHILE c( frontIsClear c) w(
                IF c( not c( frontIsClear c) c) i(
                    pickMarker i)
                move w)
        w)
    i)
m)
```

**Level 2** :

```
DEF run m(
    IFELSE c( leftIsClear c) i(
        WHILE c( leftIsClear c) w(
            move
            IF c( rightIsClear c) i(
                turnLeft
                move
                turnLeft
                turnLeft
                move
                putMarker i)
            WHILE c( frontIsClear c) w(
                move w)
        w)
    i)
    ELSE
        e( WHILE c( leftIsClear c) w(
            REPEAT R=4
                r( move r) w)
        e)
m)
```

### H.2.4 SEEDER

**Level 1** :

```
DEF run m(
    turnLeft
    WHILE c( noMarkersPresent c) w(
        putMarker
        REPEAT R=10 r(
            move r)
        REPEAT R=5 r(
            WHILE c( markersPresent c) w(
                IF c( noMarkersPresent c) i(
                    pickMarker i)
                turnLeft
                move
                turnRight w)
            pickMarker r)
    w)
    WHILE c( frontIsClear c) w(
        turnLeft w)
    m)
```

**Level 2** :

```
DEF run m(
    turnLeft
    WHILE c( noMarkersPresent c) w(
        putMarker
        REPEAT R=10 r(
            move r)
        REPEAT R=5 r(
            WHILE c( markersPresent c) w(
                IF c( noMarkersPresent c) i(
                    REPEAT R=5 r(
                        turnLeft pickMarker r) i)
                turnLeft
                move
                turnRight w)
            pickMarker r)
    w)
    WHILE c( frontIsClear c) w(
        IF c( not c( frontIsClear c) c) i(
            putMarker i)
        turnLeft w)
    m)
```

## H.2.5 TopOff

**Level 1** :

```
DEF run m(
    move
    REPEAT R=10 r(
        IF c( markersPresent c) i(
            WHILE c( rightIsClear c) w(
                WHILE c( noMarkersPresent c) w(
                    pickMarker w)
                move
                move w)
            putMarker
            WHILE c( not c( leftIsClear c) c) w(
                turnLeft
                move
                turnLeft
                move
                turnRight w)
        i)
        move r)
    m)
```

**Level 2** :

```
DEF run m(
    move
    REPEAT R=10 r(
        IF c( markersPresent c) i(
            WHILE c( rightIsClear c) w(
                WHILE c( noMarkersPresent c) w(
                    pickMarker
                    REPEAT R=5 r(
                        turnRight r) w)
                move
                move w)
            putMarker
            WHILE c( not c( leftIsClear c) c) w(
                turnLeft
                move
                turnLeft
                move
                turnRight w)
        i)
        move r)
    m)
```

OVERVIEW OF PROMPTS

The prompts for explanation, reconstruction, and verification of Programmatic Policies for Mi-
croRTS and Karel the Robot are detailed in the subsequent sections. Karel programs are written in
the Karel the Robot domain-specific language, while the MicroRTS ones are written in MicroLan-
guage. Our prompt includes a repeated component related to the domain description and DSL. We
provided this in full in the first prompt and, in subsequent prompts, refer to it simply as "**DSL**" to
avoid repetition.

# I KAREL PROMPTS

## I.1 EXPLAINER PROMPT

Karel the Robot is a programming environment where a robot named Karel operates in a grid-based world. Karel can perform tasks such as moving in four directions, picking up and placing markers, and checking for walls. The environment is designed to help learners grasp programming concepts through tasks that involve navigation, item manipulation, and simple logic. Tasks vary in complexity, ranging from straightforward activities like moving from one location to another, to more intricate challenges that require the use of loops and conditionals to solve mazes, manage markers in specific patterns, or adapt to dynamic changes in the environment. Here is the description of a task that we are interested in: This hands-on approach facilitates learning by doing, allowing users to directly apply programming constructs to visually evident problems and scenarios.

Below, the DSL for Karel the Robot is written as a context-free grammar (CFG). This grammar is used to write programs in Karel DSL.

```
<CFG>
Program p -> DEF run m( s  m)
Statement s -> WHILE c( b c) w( s w) |  IF c( b c) i( s i) |
IFELSE c( b c) i( s i) ELSE e( s e)  |  REPEAT R=n  r( s r) |
s;s | a
Condition b -> h |  not c( h c)
Number  n -> 0..19
Perception h -> frontIsClear | leftIsClear | rightIsClear |
markersPresent | noMarkersPresent
Action a -> move | turnLeft | turnRight |  putMarker | pickMarker
</CFG>
```

The code for playing this game is written in Karel specific DSL. This DSL describes the control flows as well as the perception and actions of the Karel agent. Actions including `move`, `turnRight`, and `putMarker` define how the agent can interact with the environment. Perceptions, such as `frontIsClear` and `markerPresent`, formulate how the agent observes the environment. Control flows, e.g., `if`, `else`, `while`, enable representing divergent and repetitive behaviors. Furthermore, Boolean and logical operators like `and`, `or`, and `not` allow for composing more intricate conditions.

For instance, this is an example of a program synthesized using this grammar in Karel DSL: `DEF run m(IF c( markersPresent c) i( pickMarker move i) m)`

The way this program is generated is as following: at the root of the grammar, we have `p`, which represents the production rule `Program p -> DEF run m( s m)`. The non-terminal `s` is transformed with the production rule `s -> IF c( b c) i( s i)`. The Boolean expression of the if statement is given by `b -> h` and `h -> markersPresent`. Finally, the body of the if statement is given by `s -> s;s`, which branches into `s -> pickMarker` and `s -> move`. Now, given the above information, try to understand and relate attributes and functions explained above in the context of Karel playing strategies.

Alright, let's take a look at program P in which I want you to write an explanation for: (this task is performed in a 8 * 8 grid).

*Program P*

The following 7 are some guidelines for writing an explanation for this program:

1. Please write a high-level explanation and do not explain the code line by line.

2. Try to understand what is happening in the code and explain it in natural language for someone who wants to know how Karel the Robot will behave using this program.

3. Write the explanation inside `<explanation></explanation>` tag.

4. DO NOT use any quotation marks in writing the explanation.

5. At the end of the explanation, write the overall goal of the program as well. (include it inside the `<explanation>` tag)

6. Do not explain it in a way that gives a clear hint on the implementation. For example, if you say: Karel enters into a cycle it implies that there is a for or while loop. So, avoid any kind of these explanations.

7. You MUST NOT talk in terms of for-loops and programming language jargon as people not familiar with programming will not understand the explanation. Also, you MUST not use any 'nest' or 'nested' words as they are related to programming elements. You should talk in terms of priorities of the actions, as you would explain the program to a person.

So, following the instructions, can you provide a high-level explanation of the provided program that another instance of LLM can rewrite this program from that summary?

## I.2 RECONSTRUCTOR PROMPT

$<< DSL >>$ (same as the explanations provided at the beginning of the Explainer)

Now, you have the background you need to know! Now, given the above information, try to understand and relate attributes and functions explained above in the context of Karel playing strategies. Here is a summary of a program generated by a large language model. Let's Call it 'LLM-Explanation' which was generated as an explanation of a program for playing a specific task in Karel:

*Explanation E*

The following 13 are some guidelines for writing a program in Karel:

1. There is NO NEED TO write classes, initiate objects.

2. Write all the code in one section. Don't write any functions. Also, don't leave any part non-implemented.

3. Write only the pseudocode inside `<strategy></strategy>` tag. Also, write the whole code in a single line.

4. Don't use any external information from any resources. Just rely on the explanation and the prompt provided to generate the code.

5. Please carefully distinguish between the reconstruction of IF and WHILE statements, as they can be easily confused. Pay close attention to the explanation provided to identify the differences between them.

6. There is no empty character. We don't have things like: ELSE e( e). Don't use undefined things!!

7. At the end, double check your code with the DSL and grammar. Be careful not to add any garbage code!

8. Be careful about the spaces too. For instance these are all wrong (h is an example of a syntax): c(hc) or this c( hc), or this c ( h c) or c( h c ). Instead, it should be c( h c). So, when you write something, check if all the spacing rules are being followed.

9. A common mistake is this: c(not c(h c)). The correct version is: c( not c( h c) c). Be careful about spaces.

10. Don't forget the space after the parentheses. For instance: m( WHILE ...

11. When you use ELSE, the corresponding IF should be in the form of IFELSE.

12. Only write the code, no explanation is needed. Also, write all the code in ONE line.

13. Once again, remember to implement everything. Don't leave any helper function unimplemented.

Your tasks are the following 6:

1. Understand the meaning of all functions and variables and try to relate them in the context of Karel programs.

2. Given the instructions on how to write a program and the explanation from the large language model ('LLM-Explanation') provided earlier, can you write down the program encoded in the explanation and reconstruct the program in the Karel scripting language?

3. Make sure you only use the functions, and attributes that I introduced earlier (don't use any extra keywords or signs such as THEN; UNTIL etc.).

4. Check the generated code and make sure you are following Karel DSL and grammar (make sure it is generated from the DSL). One of the common mistakes is to use WHILE/IF/IFELSE/ELSE statements without using their correct notations. For instance, the correct form is 'while c( b c) w( s w)'.

5. Check for the parentheses and spaces, they are really important in this language.

6. Don't use any assumptions of your own except for the ones that I noted to write the code.

## I.3 VERIFIER PROMPT

$<< DSL >>$ (same as the explanations provided at the beginning of the Explainer)

Now, you have the background you need to know!
I have a program P which is written in Karel the Robot language. I'm asking the first instance of an LLM for a high-level explanation without any programming jargon about the program so that another instance of LLM can regenerate the code using that explanation. The first LLM gave me this:

*Explanation E*

Now, I want to know if the explanation is using any computer programming jargon or provides step by step or line by line instructions to reconstruct the program? (Answer with yes or no first and then explain why)

# J MICRORTS PROMPTS

## J.1 EXPLAINER PROMPT

MicroRTS (ONTANÓN, 2013) is an implementation of a real-time strategy game, played between two players. Each player controls a set of units of different types.

- Worker units can:
    1. collect resources
    2. build structures (Barracks and Bases)
    3. attack opponent units
- Barracks and Bases:
    1. Barracks can train combat units. (they can produce Light, Heavy or Ranged units)
    2. Bases can train the Workers. (they can only produce Workers)
    3. They can neither attack opponents units nor move.
- Combat units:
    1. can be of type Light, Heavy, or Ranged.
    2. These units differ in:

- how long they survive a battle
- how much damage they can inflict to opponent units
- how close they need to be from opponent units to attack them.

3. They can attack the oppponent units.

- Resource units:

1. The source of resources, doesn't belong to any player.
2. These units cannot execute any actions.
3. When the number of resources left reaches 0, the unit disappears. (It only has one parameter: resources left.)

Actions are deterministic and there is no hidden information in MicroRTS. A match is played on a map and each map might require a different strategy for defeating the opponent.

This CFG allows nested loops and conditionals. It contains several Boolean functions (B) and command-oriented functions (C) that provide either information about the current state of the game or commands for the ally units. The following describes a scripting language for playing MicroRTS. The Boolean functions are described below:

1. u.hasNumberOfUnits(T, N): Checks if the ally player has N units of type T.

2. u.opponentHasNumberOfUnits(T, N): Checks if the opponent player has N units of type T.

3. u.hasLessNumberOfUnits(T, N): Checks if the ally player has less than N units of type T.

4. u.haveQtdUnitsAttacking(N): Checks if the ally player has N units attacking the opponent.

5. u.hasUnitWithinDistanceFromOpponent(N): Checks if the ally player has a unit within a distance N from a opponent's unit.

6. u.hasNumberOfWorkersHarvesting(N): Checks if the ally player has N units of type Worker harvesting resources.

7. u.is_Type(T): Checks if a unit is an instance of type T.

8. u.isBuilder(): Checks if a unit is of type Worker.

9. u.canAttack(): Checks if a unit can attack.

10. u.hasUnitThatKillsInOneAttack(): Checks if the ally player has a unit that kills an opponent's unit with one attack action.

11. u.opponentHasUnitThatKillsUnitInOneAttack(): Checks if the opponent player has a unit that kills an ally's unit with one attack action. v u.hasUnitInOpponentRange(): Checks if an unit of the ally player is within attack range of an opponent's unit.

12. u.opponentHasUnitInPlayerRange(): Checks if an unit of the opponent player is within attack range of an ally's unit.

13. u.canHarvest(): Checks if a unit can harvest resources.

The Command functions are described below. These functions assign actions to units.

1. u.build(T, D, N): Builds N units of type T on a cell located on the D direction of the unit.

2. u.train(T, D, N): Trains N units of type T on a cell located on the D direction of the structure responsible for training them.

3. u.moveToUnit(T_p, O_p): Commands a unit to move towards the player T_p following a criterion O_p.

4. u.attack(O_p): Sends N Worker units to harvest resources.

5. u.harvest(N): Sends N Worker units to harvest resources.

6. u.idle(): Commands a unit to stay idle and attack if an opponent unit comes within its attack range.

7. u.moveAway(): Commands a unit to move in the opposite direction of the player's base. 'T' represents the types a unit can assume. 'N' is a set of integers. 'D' represents the directions available used in action functions. 'O_p' is a set of criteria to select an opponent unit based on their current state. 'T_p' represents the set of target players. 'e' is an empty block which means doing nothing.

The for loops in this scripting language iterate over all units and the instructions inside the for loops attempt to assign actions to each of these units. For example, for (Unit u) u.build(Barracks, EnemyDir, 8)

The snippet above will assign a build action to unit u. Note that the only unit that can build is Worker. In the for loop above, if u is Ranged, for example, then the instruction u.build(Barracks,EnemyDir,8) will be ignored. Once an action is assigned to a unit, it cannot be changed. That is why the for loops offer a priority scheme to the actions. The way that for loop are organized is perhaps the most important feature of this language. The program always has the form: for (Unit u) ... With the possibility of adding nested for-loops that go through all units. The parameter in each instruction limits the amount of the thing that is trained or build. For example, u.build(Barracks, EnemyDir, 8) limits to at most 8 Barracks. If the player already has 8 barracks, then this instruction will be ignored. Now, you have the background you need to know!

Now, given the above information, first try to understand the meanings of all the boolean (B) and command (C) functions from above and try to relate them in the context of microRTS playing strategies.

Alright, let's take a look at program P in which I want you to write an explanation for:

*Program P*

The following 7 are some guidelines for writing an explanation for this strategy:

1. Please write a high-level explanation and do not explain the code line by line but try to include numbers and unit names in your natural language explanation.

2. Try to understand what is happening in the code and explain it in natural language for someone who wants to know how to play MicroRTS using this strategy.

3. Write the explanation inside '$< explanation >< /explanation >$' tag.

4. DON'T USE any quotation marks in writing the explanation.

5. At the end of the explanation, write the overall goal of the strategy as well. (include it inside the explanation tag)

6. Don't forget to mention the numbers but in a natural language way.

7. The for-loops offer a hierarchy that determines the priority of the actions.

The list below specifies the different priorities for actions one can obtain with the for-loops (from highest to lowest priority).

- Actions inserted in nested for-loops at the top of the program receives the highest priority.

- Actions inserted in nested for-loops that appear later in the program have higher priority than actions that appear outside a nested for loop.

- Actions outside the nested for-loops that appear earlier in the program have higher priority than actions outside for-loops that appear later in the program.

The most important part of the explanation is to be clear with respect to the priority of the actions. That is, what is the action with highest priority, which one follows that one, and so on. You MUST NOT talk in terms of for-loops and programming language jargon as people not familiar with programming will not understand the explanation. Also, you MUST not use any 'nest' or 'nested' words as they are related to programming elements. You should talk in terms of priorities of the actions, as you would explain the strategy to a gamer.

For example, if an action is within a nested for loop, then you need to say that the priority to this action is the highest. If an action isn't within a nested for loop, then you must say that the priority isn't the highest.

So, following the instructions, can you provide a high-level explanation of the provided program that another instance of LLM can rewrite this program from that summary? Remember to verify the priority of the actions, they should be well explained in your text (e.g., this action has the highest priority, this has the second highest and so on).

## J.2 RECONSTRUCTOR PROMPT

$<< DSL >>$ (same as the explanations provided at the beginning of the Explainer)

Now, you have the background you need to know!
Here is a summary of a programmatic strategy generated by a large language model. Let's Call it 'LLM-Explanation' which was generated as an explanation of a strategic program for playing MicroRTS.

*Explanataion E*

The following 7 are some guidelines for writing a program in MicroRTS:

1. There is NO NEED TO write classes, initiate objects such as Unit, Worker, etc.

2. You should NOT WRITE any comments in the code. (A comment is written in this format //comment, so avoid it!)

3. Use curly braces like C/C++/Java while writing any 'for' or 'if' or 'if-else' block. Start the curly braces in the same line of the block.

4. Do not write 'else if(B) {' block. Write 'else { if(B) {...}}' instead.

5. The format of the generated code must be the same as the example provided earlier. (not the content, the just the general structure)

6. Write only the pseudocode inside '$< strategy >< /strategy >$' tag.

7. The strategy should be written inside one or several 'for' blocks.

Your tasks are the following 11:

1. Understand the meanings of all the boolean (B) and command (C) functions from above and try to relate them in the context of microRTS playing strategies.

2. Given the instructions on how to write a program and the explanation from the large language model('LLM-Explanation') provided earlier, can you write down the strategy encoded in the explanation and reconstruct the program in the MicroRTS scripting language? (Please only use the language provided)

3. You must not use any symbols (for example: &&, ||, etc.) outside this given CFG. You have to strictly follow this CFG while writing the pseudocode.

4. Look carefully, the methods of non-terminal symbols B and C have prefixes 'u.' in the examples since they are methods of the object 'Unit u'. You also need to follow the patterns of the example provided earlier.

5. Write only the pseudocode inside '$< strategy >< /strategy >$' tag.

6. Do not write unnecessary symbols of the CFG such as, '$- >$', '$- >$', etc.

7. When you encounter parentheses in an expression or code, their opening and closing positions are crucial as they indicate the inclusion of some statements within others. The most important feature of this language is how the nested for-loops work and where they start and finish. The for-loops offer an hierarchy that determines the priority of the actions. The list below specifies the different priorities for actions one can obtain with the for-loops (from highest to lowest priority).

8. Actions inserted in nested for-loops at the top of the program receives the highest priority.

9. Actions inserted in nested for-loops that appear later in the program have higher priority than actions that appear outside a nested for loop.

10. Actions outside the nested for-loops that appear earlier in the program have higher priority than actions outside for-loops that appear later in the program.

11. Check the pseudocode and ensure it does not violate the rules of the CFG or the guidelines of writing the strategy.

IMPORTANT:

- Conditional structures such as if-statements are rarely needed. Use an if-statement only if you are sure that you need them to implement the strategy. For example, you should NOT use if-statements to ensure that a number of units is trained as these numbers are already handled by the action functions.
- For efficiency reasons, your program should have at most 2 levels of nested loops.
- Double check whether the priorities of the actions are matching with the nested for-loop structure of your program.
- The instructions with highest priority should be placed in innermost loops. If you aren't sure about an action, then leave it in the main for-loop.

## J.3    VERIFIER PROMPT

$<< DSL >>$ (same as the explanations provided at the beginning of the Explainer)

Now, you have the background you need to know!

I have a program P which is written in MicroRTS language. Then, I'm asking the first instance of an LLM for a high-level explanation without any programming jargon about the program so that another instance of LLM can regenerate the code using that explanation. The first LLM gave me this explanation:

*Explanation E*

Now, I want to know if the explanation is using any computer programming jargon or provides step by step or line by line instructions to reconstruct the program? (Answer with yes or no first and then explain why)

# K    LLM BASELINE PROMPT

Prompt we use with the LLM Baseline.

$<< DSL >>$ (same as the explanations provided at the beginning of the Explainer)

Below is a list of 4 programs written in the described scripting language. Please assign an interpretability score to each program, ranging from 0 to 1, where 1 indicates the program is highly interpretable and 0 indicates it is not interpretable at all. When assessing interpretability, consider not only the clarity of logic, simplicity of structure, and readability of the code but also how clearly the behavior of the program is understood, especially in terms of decision-making and strategic planning within the context. This is the list of N programs:

P1:
P2:
.
.
PN:

After scoring, display the results as follows:

- First line: List the scores for each program, separated by a dash (-).
- Second line: List the programs in order from most to least interpretable, starting with P1 through PN.

> Note that you should only output those two lines without any extra characters. (just the scores and the ranking next line)

## L SIGNIFICANT/NOT SIGNIFICANT TABLES

### L.1 KAREL THE ROBOT

| Policy | StairClimber | Maze | FourCorners | Seeder | TopOff |
|--------|--------------|------|-------------|--------|--------|
| StairClimber | — | NS | S | S | S |
| Maze | NS | — | S | S | S |
| FourCorners | S | S | — | S | NS |
| Seeder | S | S | S | — | NS |
| TopOff | S | S | NS | NS | — |

Table 8: Pairwise comparison for Karel the Robot (I-Score). S = Significant, NS = Not Significant.

| Policy | StairClimber | Maze | FourCorners | Seeder | TopOff |
|--------|--------------|------|-------------|--------|--------|
| StairClimber | — | NS | S | NS | S |
| Maze | NS | — | NS | NS | S |
| FourCorners | S | NS | — | NS | S |
| Seeder | NS | NS | NS | — | S |
| TopOff | S | S | S | S | — |

Table 9: Pairwise comparison for Karel the Robot (V-Score). S = Significant, NS = Not Significant.

### L.2 MICRORTS

| Policy Type | $\pi_1$ (Original) | $\pi_2$ (Original) | $\pi_1$ (Obfuscated) | $\pi_2$ (Obfuscated) |
|-------------|--------------------|--------------------|----------------------|----------------------|
| $\pi_1$ (Original) | — | NS | S | S |
| $\pi_2$ (Original) | NS | — | NS | S |
| $\pi_1$ (Obfuscated) | S | NS | — | NS |
| $\pi_2$ (Obfuscated) | S | S | NS | — |

Table 10: Pairwise comparison for MicroRTS (I-Score). S = Significant, NS = Not Significant.

| Policy Type | $\pi_1$ (Original) | $\pi_2$ (Original) | $\pi_1$ (Obfuscated) | $\pi_2$ (Obfuscated) |
|-------------|--------------------|--------------------|----------------------|----------------------|
| $\pi_1$ (Original) | — | NS | NS | NS |
| $\pi_2$ (Original) | NS | — | NS | NS |
| $\pi_1$ (Obfuscated) | NS | NS | — | NS |
| $\pi_2$ (Obfuscated) | NS | NS | NS | — |

Table 11: Pairwise comparison for MicroRTS (V-Score). S = Significant, NS = Not Significant.

