# OpenReview forum: "Assessing the Interpretability of Programmatic Policies using Large Language Models"
_ICLR.cc/2025/Conference — Submitted to ICLR 2025_

### Official Review · Reviewer_deQy · 2024-10-28

**Soundness:** 3
**Presentation:** 3
**Contribution:** 3
**Rating:** 5
**Confidence:** 2

**Summary:**

The paper proposes the LLM-based Interpretability score as a method for evaluating the interpretability of programmatic policies. The approach leverages LLMs to predict human judgments of interpretability. Experiments are conducted in two domains, MicroRTS and Karel the Robot, with the goal of demonstrating LINT's effectiveness in assessing how easily humans can understand policies.

**Strengths:**

- The idea of using LLMs to evaluate policy interpretability is creative and shows promise in leveraging existing language models for new applications.

- Interdisciplinary Relevance: The integration of LLMs into policy interpretability assessment is an exciting intersection of interpretability research and natural language processing, which could be beneficial to both communities.

- The paper is well-structured and the experiments are explained in detail, which makes it easy to follow the motivation and method behind LINT.

**Weaknesses:**

- The LINT score's novelty would be more compelling if it were benchmarked against existing interpretability metrics. The absence of such comparisons leaves questions about its relative performance. I am confused about why the authors propose using LLMs for evaluation. What are the advantages compared to previous methods? I believe that LLMs are essentially black boxes, which could potentially increase the lack of explainability.

- The user studies are conducted with small sample sizes (n = 60 for MicroRTS, n = 33 for Karel). This limits the statistical power and generalizability of the findings.

- The experiments are confined to just two simple domains, which may not translate well to more complex or diverse real-world scenarios. Broader testing would help in validating the generalizability of LINT.

- The paper does not adequately address how biases inherent in LLMs might affect the validity of LINT as an interpretability measure. A deeper analysis of LLM bias would help in evaluating the reliability of this method.

- LINT struggles to differentiate policies with small differences in interpretability. Enhancing this sensitivity is crucial for practical applications where subtle interpretability changes matter.

**Questions:**

Please refer to weaknesses.

---

> ### Author Response · Authors · 2024-11-17
> **Thank you**
>
> **Add baselines to the experiments**
>
> **Answer:** Thank you for suggesting the inclusion of baselines. We have added two baselines to the paper. In the first baseline, we use an LLM to give an interpretability score to a set of programs passed as input. We use prompts similar to the ones we use with LINT, but we bypass the “natural language bottleneck” LINT uses. The results of this baseline appear in the last column of Tables 4 and 5 of the revised version. While the LINT score presented correlations of 0.447 with I-Score and 0.204 with V-Score for the Karel programs, the LLM Baseline presented correlations of 0.01 and 0.02 with I and V-Scores, respectively, for the same set of programs. As for MicroRTS, the correlations for LINT were 0.278 and 0.126, while for the LLM Baseline were -0.16 and -0.07. This baseline shows that LINT is an improvement over the use of LLMs without the natural language bottleneck. These results are detailed in Section 6.2 of the revised manuscript.
>
> In the second baseline, we consider simple structural metrics such as number of lines of code, number of loops and conditionals. In Appendix E of the revised manuscript (Table 7) we show that none of these metrics can capture interpretability of a set of programs, while the LINT score does.
>
> **Limitations of sample size in user studies**
>
> **Answer:** We agree that it limits the statistical power, but the sample size in our study was sufficient to obtain statistically significant results in many of our comparisons (the small p-values we presented in the paper). What this means is that even if we increased the sample size by a large number, the conclusions of the studies would very likely remain the same.
>
> **Domain limitations in LINT experiments**
>
> **Answer:** We have included results on obfuscated C programs in Appendix E of the revised version, to offer yet another data point for LINT.
>
> **Impact of LLM biases on LINT reliability**
>
> **Answer:** This is a great point, and something we will have to study in future research.
>
> **Sensitivity of LINT to subtle differences in interpretability**
>
> **Answer:** We did not design LINT with the subtle differences in interpretability in mind. This is for two reasons. First, it is very difficult to flag these subtle differences. Second, they do not matter in practice. If a human cannot distinguish between two similarly interpretable programs, then this difference will not matter in practice. Please let us know if you have a particular application in mind where subtle differences would matter; we would love to hear more about it.

---

> > ### Comment · Reviewer_deQy · 2024-11-25
> >
> > Thanks for your kind response. I would prefer to retain my original scores.
> >
> > If subtle interpretability differences become a concern for certain high-stakes applications (e.g., medical AI or critical systems), the LINT score could evolve to include additional metrics or techniques to detect these nuances. Althogh the human cannot distinguish between two similarly interpretable programs, the difference also important in sometime.
> >
> > Maybe the authors can use LLMs to analyze specific aspects of interpretability (e.g., modularity, simplicity, alignment with human reasoning). Subtle differences could be flagged in the detailed score, while the high-level score remains general for practical use. Another approach is to use methods that amplify this difference, such as incorporating classical statistical models.

---

> > > ### Author Response · Authors · 2024-11-25
> > >
> > > Thank you for reading our rebuttal and for writing a response to it—we appreciate your engagement.
> > >
> > > We love your suggestions that future work break the LINT score down into multiple dimensions. We did not think of it before, and we believe they represent wonderful research directions the community will be able to pursue once our paper is published.
> > >
> > > We would like to clarify a possible misunderstanding, though. What you are calling subtle differences in your suggested framework might be different from the subtle differences we noted in our experiments. As a hypothetical example from your framework, consider two programs: A and B. Programs A and B are both highly modular, but A does not align with human reasoning as much as B. You are calling this difference subtle because it occurs in only one interpretability dimension (human reasoning alignment).
> > >
> > > What we have been calling subtle is different. If the difference in “human alignment score” between A and B is too small to be perceived by a human, then this difference is subtle and not important. If the difference was large enough so that humans can actually perceive it, then this is what we want our system to capture.
> > >
> > > Having said that, let us consider the case that we do need to design a system that captures these small differences. Since humans cannot distinguish between A and B if the difference between them is too small, how would we find the ground truth to evaluate this system? Such a system is not only very difficult to design, but impossible to evaluate.
> > >
> > > We hope this explains why we are not concerned with small differences in interpretability scores, and why they should be disregarded.

---

### Official Review · Reviewer_gZii · 2024-10-30

**Soundness:** 2
**Presentation:** 2
**Contribution:** 3
**Rating:** 6
**Confidence:** 4

**Summary:**

This paper introduces the LINT score, a metric for measuring the interpretability of a programmatic reinforcement learning policy. The idea is based on the intuition that the more interpretable a program is, the easier it is to describe it in natural language. The authors first use an LLM to explain a program in natural language, then ask another LLM to reconstruct the program from the explanation, measuring the similarity between the two versions. Their framework also includes a verifier to ensure that the explanation does not provide line-by-line details for program reconstruction.

They evaluated their method on MicroRTS and Karel the Robot, demonstrating that the more obfuscated the code, the less similar the two versions of the program will be. They also conducted a user study to test whether the proposed metric aligns with human preference.

**Strengths:**

Recent advancements in programmatic reinforcement learning (PRL) primarily focus on improving performance, despite interpretability being a foundational goal of PRL. This makes the motivation for measuring how interpretable a programmatic policy is well-founded, as a program’s interpretability can naturally vary between its language and underlying code. The approach of asking an LLM to explain and then reconstruct the program is both innovative and intuitively sound.

**Weaknesses:**

**The problems of evaluating the results on Karel.** In the Karel the Robot environment, there should be ten tasks, including six from Karel and four from Karel-Hard. What is the reason for selecting the five tasks in the experimental results and not the others? I see you chose Seeder as one of the test beds from Karel-Hard, so why didn’t you include the other tasks as well? Another issue is that there is only one policy for each task, which seems insufficient for a comprehensive evaluation of the results. Additionally, the action metric appears unclear. In Karel the Robot, the action an agent performs depends on both the grid it occupies and the code blocks it is executing. In other words, the agent may perform different actions even when on the same grid. How do you define "state" in this context?

**The experimental evidence is hard to confirm.** In Table 3, the paper lists only the average results for all policies, without including the per-task results. Regarding the level of obfuscation, it appears that you did not specify how you inserted the useless code blocks. Was this done through a rule-based method or with human involvement? Additionally, what are the standards for levels 1 and 2? In Line 168, you mentioned that you only invoked the verifier once, so how can you claim that "all approved explanations satisfied the constraints"?

**Questions:**

- As you mentioned in Section 4, the time spent pondering may affect the results. So what is the time limit for users to answer the questions?
- Given that users rate program interpretability on a scale from 1 to 5, can you instruct the LLM to answer the same questions and calculate the correlation with users' responses?
- I noticed that you included the prompts for Karel in the appendix, but I’m still curious whether the LLM has access to task information when generating both the explanation and the reconstructed program. It seems intuitive that understanding a program would be easier when the purpose of the code blocks is provided.

---

> ### Author Response · Authors · 2024-11-17
> **Thank you**
>
> **There should be ten tasks, including six from Karel and four from Karel-Hard. What is the reason for selecting the five tasks in the experimental results and not the others? I see you chose Seeder as one of the test beds from Karel-Hard, so why didn’t you include the other tasks as well? Another issue is that there is only one policy for each task, which seems insufficient for a comprehensive evaluation of the results.**
>
> **Answer:** We used only 5 tasks due to limitations that are common to a user study. That is, we have to be mindful of the participants of the study, since they cannot evaluate a large number of programs. The ethics board of our institution would naturally not approve such a lengthy experiment. We are also limited by the number of participants we can hire, as we depend on people with a programming background to volunteer to take part in the study. We chose the 5 programs because they offered a varied level of difficulty for the participants. We respectfully disagree that the number of programs used was insufficient for a comprehensive evaluation of the method, as clearly captured by the statistically significant results across two entirely different domains. We agree that having more programs and more domains is always beneficial. Given the cost of running user studies, this does not have to be a one-group endeavor, but a community effort.
>
> **Clarifying the action metric in Karel**
>
> **Answer:** We save the states in which the original program is evaluated and evaluate the reconstructed program in exactly the same state. For example, in Karel, the state specifies exactly how the grid looks and the position of Karel. For the action metric, we compute the fraction of actions chosen by the original program p that match the actions chosen by the reconstructed program p’. To do so, we fix the states and compare the actions generated in that state by running two different code blocks (p and p’).
>
> **The experimental evidence is hard to confirm..**
>
> **Answer:** We averaged the results in Table 3 to save space, as the individual results do not provide new insights.
>
> **Methodology for code obfuscation**
>
> **Answer:** We had a set of fixed useless code that we added to each program. It was done with human involvement but can also be scripted to be done automatically.
>
> **Additionally, what are the standards for levels 1 and 2? In Line 168, you mentioned that you only invoked the verifier once, so how can you claim that "all approved explanations satisfied the constraints"?**
>
> **Answer:** We manually checked all executions of the system, and we agreed with the verifier that the constraints were satisfied.
>
> **Time limits in user studies**
>
> **Answer:** We don’t have a time limit in our user study as adding it could add a stress factor to the participants. In addition to being stressful for the participants, a time limit could increase the amount of noise in the data we collected. Our evaluation is based on the premise that participants could take as much time as they needed to answer the questions.
>
> **Given that users rate program interpretability on a scale from 1 to 5, can you instruct the LLM to answer the same questions and calculate the correlation with users' responses?**
>
> **Answer:** Great suggestion. Following your suggestion, we added a baseline that does exactly that (LLM Baseline). We use prompts similar to the ones we use with LINT, but we bypass the “natural language bottleneck” LINT uses. The results of this baseline appear in the last column of Tables 4 and 5 of the revised version. While the LINT score presented correlations of 0.447 with I-Score and 0.204 with V-Score for the Karel programs, the LLM Baseline presented correlations of 0.01 and 0.02 with I and V-Scores, respectively, for the same set of programs. As for MicroRTS, the correlations for LINT were 0.278 and 0.126, while for the LLM Baseline were -0.16 and -0.07. This baseline shows that LINT is an improvement over the use of LLMs without the natural language bottleneck. These results are detailed in Section 6.2 of the revised manuscript.
>
> **LLM access to task information**
>
> **Answer:** We only provide information about the problem domain and not the task. This is because we wanted the LLM to use only the information coming from the program (explanation) and from the natural language description (reconstruction). Otherwise, the information we provided in the prompt could create false positives, where the program is not interpretable, yet the LLM uses information from the prompt to flag it as interpretable.

---

> > ### Comment · Reviewer_gZii · 2024-11-25
> > **Thank you for your detailed responses (score raised)**
> >
> > Thank you for your detailed clarifications. I have raised my score according to it.
> >
> > I think the motivation for measuring the interpretability of a PRL policy is reasonable and should gain more attention.
> >
> > The thing I find most interesting is the LLM baseline, it's surprising that the LLM performs that badly when aligning with the human preference. This finding greatly enhances my confidence in the reported result.

---

> > > ### Author Response · Authors · 2024-11-25
> > >
> > > Thank you for reading our rebuttal and acknowledging our work by increasing your score. Your suggestion of adding this baseline was quite valuable as the added results support the use of the natural language bottleneck to measure interpretability.

---

### Official Review · Reviewer_n7Uq · 2024-11-02

**Soundness:** 3
**Presentation:** 3
**Contribution:** 2
**Rating:** 3
**Confidence:** 4

**Summary:**

This authors propose a metric called LINT score. This metric leverages LLMs to evaluate the interpretability of programmatic policies used in sequential decision-making tasks by reconstructing a policy based on its natural language description.

The proposed method includes using one LLM describe a policy's behavior in natural language, followed by a second LLM attempting to reconstruct the policy based on this description.  The similarity between the original and reconstructed policies serves as the interpretability measure. The authors hypothesize that higher similarity indicates greater interpretability. The metric was tested in two domains: MicroRTS and Karel. The experiments involve increasing the obfuscation level of policies to simulate varying degrees of interpretability, with user studies to validate the correlation between LINT scores and human perceptions of interpretability

**Strengths:**

- The paper presents the LINT score methodology with clarity, detailing each component the explainer, reconstructor, and verifier and its purpose within the interpretability assessment pipeline.
- Although initial results suggest areas for improvement, the study opens up valuable avenues for refining interpretability metrics, setting a constructive groundwork for further development in areas of robotics and RL.
- The use of varying obfuscation levels and structured user studies across domains demonstrates a good experimental design.

**Weaknesses:**

- Since the original policy is directly provided to the explainer, there’s a risk that the natural language description generated is highly specific and detailed, making it relatively straightforward for the reconstructor to recreate the policy. This dependency raises questions about whether the LINT score truly reflects interpretability or merely the ability of another LLM to follow a sufficiently descriptive prompt.
- The paper defines interpretability narrowly as the ability to reconstruct a policy from a natural language description, which may not capture the full human-centric aspects of interpretability.
- The LINT score is tested only on specific domains (MicroRTS and Karel the Robot), both of which are relatively contained environments with domain-specific languages. It’s unclear how well the LINT score would generalize to more complex or diverse environments.
- The method relies on the assumption that a policy that is interpretable should be reconstructable in a way that behaves similarly to the original policy. However, minor discrepancies in LLM-generated reconstructions, even if they don’t reflect actual interpretability, might impact the LINT score due to the stochastic nature of LLMs. Because LLMs are stochastic and may produce different outputs for the same prompt, achieving consistent LINT scores might require multiple runs, which complicates reproducibility.
- The paper relies solely on the LINT score for interpretability assessment without integrating other interpretability measures, such as feature importance or visualization techniques, that might provide a more comprehensive view of a policy's interpretability.
- Although the LINT score aims to be cost-effective compared to user studies, the computational expense of using multiple LLM instances (Explainer, Verifier, Reconstructor) may still be high, especially for larger-scale deployments.
- While I do feel this area of study is important, the methodology in this paper felt more like a pipeline of multiple LLMs strung together, which limits its methodological novelty.

**Questions:**

- Since programmatic policies provide explicit code with defined logic, wouldn’t they be inherently interpretable by examining their internal workings? Could you clarify why an additional interpretability metric, like LINT, is necessary when the policy’s behavior can ostensibly be understood directly from its code structure?
- Given that LLMs themselves are black boxes, what is the rationale behind using LLMs as proxies for interpretability? Wouldn't alternative methods, such as rule-based systems offer more transparency in the interpretability assessment?
- For variability in LINT scores across multiple runs? Did you consider using temperature or other settings to stabilize outputs?

---

> ### Author Response · Authors · 2024-11-17
> **Thank you**
>
> **The natural language description generated might be highly detailed, making it straightforward for the reconstructor to recreate the policy.  [...] raises questions about if the LINT score reflects interpretability.**
>
> **The paper defines interpretability narrowly as the ability to reconstruct a policy from a natural language description, which may not capture the full human-centric aspects of interpretability.**
>
> **Answer:** Thank you for raising these important questions. These were exactly the questions we asked ourselves once we designed LINT. The system makes several assumptions, such as the ones you wrote: does the dependency on its “natural language bottleneck” truly reflect interpretability? Does it capture the human-centric aspects of interpretability?
>
> We hypothesized positive answers to these questions and designed a user study to evaluate our hypothesis. As we showed in the paper, the empirical results we obtained supported our hypothesis.
>
> **Generality of LINT across domains**
>
> **Answer:** Following your suggestion, we have also included in Appendix E of the revised version the LINT results for C programs from the International Obfuscated C Code Contest, where C programs are written to be as non-interpretable as possible. LINT flagged the winners of the contest to be non-interpretable (score of 0.0) and their non-obfuscated counterparts to be fully interpretable (score of 1.0).
>
> **Stochasticity and reproducibility in LINT scores**
>
> **Answer:** We understand the concern around the stochasticity of the approach. However, this is a problem that is easy to solve. We can deal with the stochasticity of the model by running it several times, as we did in our experiments. The difficulty in reproducibility due to stochasticity is no different from that of stochastic algorithms in the AI/ML literature, such as neural networks.
>
> **Integration of other interpretability measures such as feature importance or visualization techniques**
>
> **Answer:** It is unclear how to use feature importance in the programmatic representations we use. The way these programmatic models work is entirely different from neural networks, for which this type of approach is suitable. Please let us know if we missed the point in your suggestion. Note that the V-Score in our study serves as a visualization tool, where the participant visualizes the behavior of the policy.
>
> **The computational expense of using multiple LLM instances (Explainer, Verifier, Reconstructor) may still be high, especially for larger-scale deployments.**
>
> **Answer:** We agree that the use of LLMs can be expensive, but they are still far cheaper than hiring professional programmers, training them on a novel language, and paying them to study and explain computer programs. While the LINT approach is deployable at large scale, even though it may be costly, the alternative approach of hiring human programmers is entirely infeasible at scale.
>
> **Methodological novelty of LINT**
>
> **Answer:** We agree LINT is a set of LLMs strung together, but in a novel and creative way. To give you an unbiased perspective, reviewer gZii wrote “The approach of asking an LLM to explain and then reconstruct the program is both innovative and intuitively sound”, and reviewer deQy wrote “The idea of using LLMs to evaluate policy interpretability is creative and shows promise in leveraging existing language models for new applications.”
>
> **Necessity of LINT for programmatic policies**
>
> **Answer:** The language in which the program is written does not determine whether the program is interpretable or not. We cannot assume that the behavior can be understood directly from the code. To make this point clear, we have added experiments with C programs from the International Obfuscated C Code Contest in Appendix E of the revised version. In this experiment we evaluate programs that solve simple programming problems, but that were designed to be non-interpretable. The LINT score correctly classifies the obfuscated programs as non-interpretable and the non-obfuscated as interpretable.
>
> **Rationale for using LLMs in interpretability assessment**
>
> **Answer:** The rationale is that LLMs are able to create a “natural language bottleneck” of the policy behavior; to our knowledge, no other system has a similar ability. Our hypothesis is that LINT's natural language bottleneck enables the measurement of interpretability. It is unclear how rule-based systems would be able to provide such a bottleneck.
>
> **For variability in LINT scores across multiple runs? Did you consider using temperature or other settings to stabilize outputs?**
>
> **Answer:** Yes, we tested different temperatures and ended up using a temperature of 0.7, which balances randomness and determinism. This information has been added to the last paragraph of Section 4. We accounted for variability by running the system multiple times, as explained in the last paragraph of Section 3.

---

> > ### Comment · Reviewer_n7Uq · 2024-11-23
> >
> > Thank you for the clarification, but I believe there’s a misunderstanding of my initial point. I did not argue that the language itself determines interpretability; rather, I highlighted that programmatic policies, by their very nature, are inherently interpretable because their internal logic is explicitly defined and accessible for analysis unlike neural networks.
> >
> > While I understand the addition of experiments with obfuscated C programs, this example represents a contrived scenario and does not reflect the typical design of programmatic policies, which are generally intended to be clear and straightforward. Obfuscation is an exception rather than the rule. Therefore, it still seems unclear why an additional interpretability metric like LINT is necessary for evaluating programmatic policies that are already interpretable by design.
> >
> > Could you clarify whether the primary motivation for LINT is specific to edge cases like obfuscation, or is it aimed at addressing a broader issue?
> >
> > Also While I agree that leveraging LLMs can be useful for interpretability assessments, I would challenge the notion that having multiple LLMs together represents a novel contribution (a bunch of such combination can be made, without saying which will work best). This design pattern has been increasingly employed in various domains.
> >
> > The innovation in such approaches typically lies in the specific use case, methodology, or insights gained. However, in the case of LINT, I still find the novelty claim questionable.

---

> > > ### Author Response · Authors · 2024-11-24
> > >
> > > Thank you for reading our rebuttal and asking follow-up questions. We appreciate your engagement and are happy to answer your questions.
> > >
> > > **Interpretability of Programmatic Policies**
> > >
> > > Thank you for your comment about the assumption that programmatic policies are inherently interpretable. This is precisely the assumption we challenge in our paper.
> > >
> > > Our reasoning is that interpretability is not binary (the program is or is not interpretable), as assumed in previous work. We hypothesized that different people may or may not be able to interpret a program, even when the language is inherently interpretable. We further hypothesized that some programs are less interpretable, while others are more interpretable. Our user study provides evidence supporting these hypotheses. For example, Table 5 compares the I-Score of the original and obfuscated programs. Despite being written in the same language—one that is supposedly inherently interpretable—the difference in interpretability is significant.
> > >
> > > Our work provides a much more nuanced perspective on the interpretability of programmatic policies than the binary assumption of previous work.
> > >
> > > **Obfuscated C Programs**
> > >
> > > We do not claim that LINT is required to assess the interpretability of obfuscated C programs; their lack of interpretability is evident. We are presenting these results to show that simple structural metrics can fail even in these obvious cases, whereas LINT is robust across all tested problems.
> > >
> > > To address your question, our interest is not in edge cases like obfuscated C programs but in the broader interpretability challenges, which we thoroughly evaluated in our user study.
> > >
> > > **Regarding Novelty**
> > >
> > > We agree with you that combining LLMs is not a novel contribution, and this is not what we are claiming. The novelty of our contribution comes from introducing a “natural language bottleneck” to measure interpretability. To our knowledge, no prior work has employed natural language for this purpose. Such a bottleneck offers a concrete interpretability metric to the subjective interpretability problem.
> > >
> > > The LLMs simply enabled us to implement this concept. The pipeline itself represents the engineering component we developed to test our hypothesis that such a bottleneck would correlate with interpretability, as validated by our user study.
> > >
> > > **Summary**
> > >
> > > We hope we have answered your questions and that it is clear to you what the novel contribution of our work is. We also hope that we managed to explain the need for an interpretability tool such as LINT, which challenges the assumption that all so-called programmatic policies are equally interpretable. If not, please do not hesitate to ask follow-up questions.

---

### Official Review · Reviewer_tPe6 · 2024-11-02

**Soundness:** 2
**Presentation:** 3
**Contribution:** 2
**Rating:** 5
**Confidence:** 4

**Summary:**

Programmatic policies for sequential decision-making problems often promise interpretability, but measuring and evaluating this promise can be challenging. User studies that assess interpretability tend to be subjective and costly. This work introduces LINT, a straightforward solution based on large language models (LLMs) to evaluate the interpretability of programmatic policies. LINT employs an LLM explainer to translate the program into natural language text, adhering to specific constraints, and then uses an LLM reconstructor to regenerate the original program from that text. If the original and reconstructed policies are similar, the original is considered interpretable. The authors demonstrate LINT’s effectiveness as an interpretability evaluator through obfuscation experiments and user studies in two domains: MicroRTS and Karel the Robot.

**Strengths:**

* The authors have pinpointed a significant challenge in the realm of programmatic policies for sequential decision-making: evaluating the interpretability of these policies. They propose a straightforward yet effective algorithm that addresses this issue, with their methodology and experimental design presented clearly. The prompts used for the LLMs and the specifics of the user studies are well detailed.
* The design choices for LINT—including the three-model setup featuring an explainer, verifier, and reconstructor, as well as the constraints for generation—are straightforward and logical. While the selected metrics, specifically the action and return heuristics, can be debated (e.g., why not consider state visitation?), they serve as defensible and practical starting points.
* The authors demonstrate a strong correlation between the LINT score and the level of obfuscation in programs, indicating that more obfuscated programs are less interpretable and thus receive lower LINT scores. This finding is promising for utilizing LLMs in code interpretability, applicable not only to programmatic policies but to any computational program.
* The user study suggests that LINT has potential as an effective tool for assessing the interpretability of programmatic policies.

**Weaknesses:**

I have three main concerns with this paper that prevent me from giving it a higher score, which I outline below:
* The primary factor that would make LINT a compelling algorithm and metric is its correlation with user interpretability. While the authors demonstrate a positive correlation through their experiments, the metrics used to assess user interpretability are unconvincing and potentially noisy. Users viewing a programmatic policy alongside the relatively straightforward domain-specific languages (DSLs) in the benchmarks might feel they understand the program and assign a high I-score, even if their understanding is superficial. Although the V-score attempts to address this, the authors acknowledge that it is a noisy metric; designing an evaluation setup with four clear options is challenging, and users can often make educated guesses. This complicates drawing conclusions from the user study results. Additionally, the scores in the table for I-scores and V-scores suggest that obfuscated programs may have scores very close to those of more interpretable ones (as seen in Table 5).
* A second challenge in evaluating LINT is the absence of baseline comparisons. Even if LINT is the first algorithm focused on evaluating the interpretability of programmatic policies, it would be beneficial to compare it against simple baselines using code heuristics—such as the number of lines, loops, and conditional clauses—to gauge interpretability. Such comparisons would strengthen the validity of LINT, particularly in light of the noisy I-scores and V-scores. Without these baselines, it raises the question of whether simple heuristic-based methods could also effectively distinguish between obfuscated and straightforward programs.
* Lastly, the LINT score itself is difficult to interpret. For instance, in Section 6.3, a policy receives a LINT score of (0.626, 0.57). This score leaves me uncertain about the interpretability of the policy; while 0.626 doesn’t seem impressive out of 1, the action score is at least bounded between 0 and 1. The return score, however, is unnormalized and unbounded, making it very domain-specific and hard to interpret. As a user, I am left unsure of how good or bad the policy truly is. In this case, the policy appears quite interpretable to me, but I would not be able to ascertain that just from the LINT score.

**Questions:**

* When I run LINT and receive scores for Action and Return, how should I interpret them? The Action score is somewhat straightforward since it ranges from 0 to 1 although there are challenges as described in the Weaknesses section. However, in grid world scenarios, multiple aliased actions for policies $\pi$ and $\pi^{'}$ can unfairly penalize the Action score. Additionally, if policy $\pi_1$ has an Action score of 0.8, $\pi_2$ has 0.7, and $\pi_3$ has 0.6, can we confidently say that $\pi_2$ is better than $\pi_3$ by the same margin that $\pi_1$ is better than $\pi_2$? Being able to make such claims would be valuable, especially when balancing interpretability against performance.
* It would also be beneficial to include a brief description of the Karel and MicroRTS environments for users who may not be familiar with them.

---

> ### Author Response · Authors · 2024-11-17
> **Thank you**
>
> **Challenges in Evaluating the Interpretability of Programs**
>
> **Answer:** Thank you for raising these concerns. The study of interpretability in machine learning is inherently noisy because it involves human factors. Although this is different from most ML research, which tends to be exact, our research is still important within the ML community. This is a challenge shared by other disciplines, such as psychology and health research, where progress is still made despite similar complexities.
>
> We designed our experiments aware of this challenge and controlled for it as much as possible. That is why we chose two, and not only one, metric. Together, they provide more information and compensate for each other's weaknesses. For this reason, we respectfully disagree that it is complicated to draw conclusions from our study. Thanks to our empirical design, we were able to collect statistically significant results that are meaningful and important to our field. Specifically, the LINT score we propose correlates with the interpretability data we collected in our study.
>
> We appreciate any suggestions you might have regarding alternative metrics that could complement or improve upon I-Score and V-Score. If you have specific recommendations or insights on how to further refine our approach, we would love to hear them.
>
> **Need for baseline comparisons**
>
> **Answer:** Thank you for suggesting the inclusion of baselines. We have added two baselines to the paper. In the first baseline, we use an LLM to give an interpretability score to a set of programs passed as input. We use prompts similar to the ones we use with LINT, but we bypass the “natural language bottleneck” LINT uses. The results of this baseline appear in the last column of Tables 4 and 5 of the revised version.
>
> While the LINT score presented correlations of 0.447 with I-Score and 0.204 with V-Score for the Karel programs, the LLM Baseline presented correlations of 0.01 and 0.02 with I and V-Scores, respectively, for the same set of programs. As for MicroRTS, the correlations for LINT were 0.278 and 0.126, while for the LLM Baseline were -0.16 and -0.07.
>
> This baseline shows that LINT is an improvement over the use of LLMs without the natural language bottleneck. These results are detailed in Section 6.2 of the revised manuscript.
>
> In the second baseline, we consider simple structural metrics such as number of lines of code, number of loops and conditionals. In Appendix E of the revised manuscript (Table 7) we show that none of these metrics can capture interpretability of a set of programs, while the LINT score does.
>
>
> **Understanding the LINT scores**
>
> **Answer:** This is a good question. The ability to interpret numbers is domain specific, as the behavior metrics are domain specific. We do not see how we could provide a general guideline in addition to “large is good” and “small is bad.” However, regardless of the intuition behind the numbers, the LINT score is helpful to compare two policies: which one is more interpretable? In this case, you are not trying to see whether the policy is “good” or “bad,” but deciding which one you prefer. Since the LINT score provides confidence intervals (as we presented in the paper), we can also have no preference if the numbers are too close, such as 0.62 and 0.57.
>
> **Understanding the action and return scores**
>
> **Answer:** The ability to interpret the scores is domain specific. Users of the system will have to decide what constitutes a sufficiently large score for them. This is similar to a hyperparameter in a learning algorithm. How much should one exchange performance for interpretability? To answer this question, the user needs to evaluate different ways of balancing the two. The important point is that the LINT score offers this possibility. Prior to our work, this was not possible, as we do not know of any other reliable interpretability metric.
>
> **Description of Karel and MicroRTS environments**
>
> **Answer:** They are available in the Appendix A.1 and A.2

---

> > ### Comment · Reviewer_tPe6 · 2024-11-23
> > **Thank you for the responses (score increased)**
> >
> > Thank you for the responses! I’ve been able to improve my score, particularly due to the additional baseline comparisons that strengthen the credibility of LINT. Below are some further comments and suggestions:
> >
> > #### **Challenges in Evaluating the Interpretability of Programs**
> > Thank you for your insightful response. I completely agree that interpretability in machine learning is not an exact science, but it remains a critical area of study. I also appreciate the authors’ thoughtful design of their experiments and metrics, keeping these inherent challenges in mind.
> > That said, I have a few suggestions for refining the I-score and V-score, and I’d be interested to hear the authors’ thoughts on these. One concern with the I-score is that participants in the user study may believe they understand a program (due to the simplicity of the DSLs in the environments) and therefore assign it a high score, even if they don’t truly grasp the logic. Perhaps a possible solution would be to have participants provide a brief written summary of what the program does and then compare that with the output of the explainer? For the V-score, it might be helpful to include multiple valid answers and offer more than one correct option out of four. This could help mitigate the guessing issue and reduce the tendency to eliminate options too easily, ultimately reducing the noise in both scores.
> >
> > #### **Need for baseline comparisons**
> > I appreciate the inclusion of the two baseline comparisons. Even though the baselines are relatively simple, they significantly enhance the credibility of LINT, and I believe this inclusion is crucial for validating the findings.
> >
> > #### **Understanding the LINT scores and action and return scores**
> > To clarify, the action score of 0.626 and return score of 0.57 that I mentioned both correspond to the same piece of code (formerly in Section 6.3, now in Appendix C). From my perspective, this program is highly interpretable, yet these scores do not seem to reflect that. One potential improvement could be to introduce a metric that tracks state visitation, such as the number of common states visited, which might offer more meaningful insights. Even with domain knowledge of Karel the Robot, it’s still difficult to interpret these scores.
> > For example, consider a grid-world task where the goal is to navigate from one corner to the diagonally opposite corner. A straightforward and interpretable policy might follow the path UP-RIGHT, but this could be reconstructed as a RIGHT-UP policy, which could yield a very low action score despite the original policy being quite interpretable. This suggests that more sophisticated metrics are needed—ones that better capture policy similarity and reflect the true interpretability of the program.

---

> > > ### Author Response · Authors · 2024-11-24
> > >
> > > Thank you for reading our rebuttal and writing such a detailed response to it—we appreciate your engagement.
> > >
> > > **Challenges in Evaluating the Interpretability of Programs**
> > >
> > > Your suggestion related to the I-Score is excellent, and we considered a similar metric when designing our study. We also considered asking participants to write a short description of the policy, as you suggested. Instead of comparing it to the LLM's description, we thought about developing a rubric to evaluate the descriptions, similar to how exams are graded. However, we decided against this approach to ensure that the study could be completed within one hour. Extending the study duration could have led to participant dropouts, leaving us without enough data points to conduct a robust statistical analysis.
> > >
> > > We do not see how the suggestion related to the V-Score could help, though. Adding more options that are correct would make it easier to guess a correct answer and thus amplify the possibility of false positives (programs that the V-Score would deem as interpretable but that are not interpretable). Please let us know if we misunderstood your V-Score suggestion.
> > >
> > > **Need for Baseline Comparisons**
> > >
> > > Thank you for checking our baseline results and increasing your score.
> > >
> > > **Understanding the LINT Scores and Action and Return Scores**
> > >
> > > Your concern is valid and goes beyond understanding the score. That is, modifying the metric could potentially improve our results. The example of “right-up” and “up-right” is great and shows how the action metric can fail. Importantly, our experiments with the action metric already allowed us to find a strong correlation between the LINT score and the user-measured interpretability. Better metrics could improve this correlation.
> > >
> > > We recognize the weaknesses of the action metric and that is why we used another metric, the return metric, which can capture some of the nuances of the program behavior that the action metric fails to capture (we use the return metric in Table 3). For example, the return metric does not suffer from the problem you described in the “right-up” and “up-right” example.
> > >
> > > **Summary**
> > >
> > > Thank you for all the great suggestions. Interestingly, with the exception of the V-Score suggestion, we had considered all of them while designing our system and study. Unfortunately, we had to decide against them, but we did so for good reasons, as we explained above.
> > >
> > > We hope we have answered your questions and that it is clear that all our design decisions and user study decisions were well thought out. Moreover, we hope that we managed to show that our work is ready for publication and that the idea of using a natural language bottleneck to approximate human interpretability can offer a promising research direction for the community. Please do not hesitate to ask follow-up questions.

---

### Official Review · Reviewer_vVdR · 2024-11-03

**Soundness:** 2
**Presentation:** 3
**Contribution:** 2
**Rating:** 3
**Confidence:** 4

**Summary:**

The author proposes a novel metric, LLM-based INTerpretability (LINT) score, which uses large-language models (LLMs) to quantify the interpretability of programmatic policies. The method leverages three LLMs to act as explainer (to generate natural language description), reconstructor (to generate policy based on language description from explainer) and verifier (to check if the explainer has not provided any unwanted information about policy). The authors validated the usability of LINT across two domains, MicroRTS and Karel the robot, demonstrating its correlation with human interpretability assessments.

**Strengths:**

- The proposed solution for quantifying the interpretability of programmatic policies is innovative but depends heavily on LLMs, which are inherently stochastic. This reliance introduces potential variability in results, but the authors address this concern by implementing measures such as multi-evaluation runs and verification steps to mitigate inaccuracies and enhance reliability.

- The authors validate the effectiveness of LINT scores across two distinct domains, MicroRTS and Karel the Robot, demonstrating a strong correlation between LINT scores and human interpretability assessments.

**Weaknesses:**

**Major:**
- **Importance of proposed method:** Currently, it is not much clear why there is a need for interpretability score for programmatic policies. From the examples shown in paper (table 1 and 2), these policies are in itself interpretable and might not warrant the need for the proposed score. Have there been any study to support the cause, authors should highlight that in introduction.
- **No theoretical guarantee:** Although the approach proposed by authors is interesting, they don't provide any theoretical guarantee as the scores they achieve is in actuality explains the policy.
- **Lack of baseline evaluation:** Although the authors conducted human study to support the method, they lack comparisons with other baseline methods. Just by looking at results from table 3 and 4 it is hard to conclude if the proposed method is an improvement or not.

**Minor:**
- Adding a brief introduction about programmatic policies in the problem definition will help in better understanding of the paper.
- The authors have described about the limitations of their approach at multiple places (lines 163 - 172 and section 4). Maybe these can be moved towards the end as a Limitations section.
- **Bad representation of data:** The letter superscript presented in table 4 and 5 for column I-score and V-score is really confusing. There should be a better way to represent that data.
- **Typo (line 300):** IFor &#8594; For

**Questions:**

1. What specific challenges in programmatic policy interpretability justify the need for a dedicated metric like LINT?

2. Are there theoretical guarantees that the LINT score reliably reflects policy interpretability?

3. How does LINT compare to other interpretability metrics, and have any baseline comparisons been considered?

---

> ### Author Response · Authors · 2024-11-17
> **Thank you**
>
> **Importance of proposed method**
>
> **Answer:** Prior to our work, papers always assumed these policies to be interpretable, by anecdotally analyzing them. Our user study contradicts this assumption and shows that some of these policies are not as interpretable as initially assumed. For example, our user study shows that the TopOff program in the Karel domain is less interpretable than the other programs evaluated (statistically significant in terms of both I and V-Scores).
>
> The results of our user study offer exactly what you are asking for the introduction: a reference that supports a deeper analysis of the interpretability of these programs.
>
>
> **No theoretical guarantee**
>
> Answer: The interpretability of programs inherently involves human factors, making it difficult to provide theoretical guarantees. This is a common limitation in this research area. Maybe we misunderstand what you mean by theoretical guarantees? If we have misunderstood your request, please feel free to clarify. We would be happy to address your concerns further.
>
> **Lack of baseline evaluation**
>
> **Answer:** Thank you for suggesting the inclusion of baselines. We have added two baselines to the paper. In the first baseline, we use an LLM to give an interpretability score to a set of programs given as input. We use prompts similar to the ones we use with LINT, but we bypass the “natural language bottleneck” LINT uses. The results of this baseline appear in the last column of Tables 4 and 5 of the revised version. The LINT score presented correlations of 0.447 with I-Score and 0.204 with V-Score for the Karel programs, while the LLM Baseline presented correlations of 0.01 and 0.02 with I and V-Scores, respectively, for the same set of programs. As for MicroRTS, the correlations for LINT were 0.278 and 0.126, while for the LLM Baseline they were -0.16 and -0.07.
>
> This baseline shows that LINT is an improvement over the use of LLMs without the natural language bottleneck. These results are detailed in Section 6.2 of the revised manuscript.
>
> In the second baseline, we consider simple structural metrics such as number of lines of code, number of loops and conditionals. In Appendix E of the revised manuscript (Table 7), we show that none of these metrics can capture interpretability of a set of programs, while the LINT score does.
>
> **Adding an introduction to programmatic policies**
>
> **Answer:** We added a brief introduction to programmatic policies, where we explain how a language is represented and how the search is performed in the space of programs the language accepts. Please see Section 2 of the revised version.
>
> **Move limitations to the end**
>
> **Answer:** We moved that section to the end of the paper. Please see Section 6 of the revised version.
>
>
> **Bad representation of data for statistical significance**
>
> **Answer:** This type of presentation is called compact letter display. We agree that it can be confusing because it conveys a quadratic relationship (all methods against all methods) in a single column of the table. We added pairwise comparison tables in Appendix L (Tables 8, 9, 10, and 11) that make it easier to compare significant and not significant relationships. Please let us know if the Appendix tables are easier to understand than the compact letter display used in the main text.
>
> **Typo (line 300): IFor → For**
>
> Thanks for your attention! Fixed.

---

> > ### Comment · Reviewer_vVdR · 2024-11-26
> >
> > Thank you for the response and addressing my queries. Now, I have better understanding for the importance of proposed method. The additional baselines, particularly the inclusion of the LLM-based and structural metric baselines, significantly improve the context and provide stronger evidence for the utility of the LINT score.
> >
> > With regards to theoretical guarantees, I acknowledge the challenge of providing them in research involving human factors. However, many established approaches in explainability and interpretability, such as LIME [1], SHAP [2], and TCAV [3], are mathematically grounded. These methods offer scores that are backed by mathematical reasoning, which enhances their credibility and interpretability. While the LINT score provides valuable insights into the interpretability of programmatic policies, it currently lacks a strong mathematical foundation to support the obtained scores. Introducing such grounding could further strengthen the metric's reliability.
> >
> > 1. Ribeiro, Marco Tulio, Sameer Singh, and Carlos Guestrin. "" Why should i trust you?" Explaining the predictions of any classifier." Proceedings of the 22nd ACM SIGKDD international conference on knowledge discovery and data mining. 2016.
> > 2. Lundberg, Scott. "A unified approach to interpreting model predictions." arXiv preprint arXiv:1705.07874 (2017).
> > 3. Kim, Been, et al. "Interpretability beyond feature attribution: Quantitative testing with concept activation vectors (tcav)." International conference on machine learning. PMLR, 2018.

---

> > > ### Author Response · Authors · 2024-11-27
> > >
> > > Thank you for reading our rebuttal and commenting on it. We also appreciate you recognizing the importance of our method and acknowledging the addition of the baselines.
> > >
> > > Thank you for explaining that by theoretical guarantees you meant mathematically grounded. Like our approach, the methods you cited (LIME, SHAP, and TCAV) do not provide theoretical guarantees. However, similarly to them, our method can be seen as mathematically grounded in that it leverages functions to map programs to natural language and back, measuring interpretability through these transformations.
> > >
> > > The concept of a method being mathematically grounded can be somewhat subjective, but we believe we understand the point you are making. For example, LIME is mathematically simple, which makes it easy to understand what it does. In contrast, we use LLMs in our pipeline, so it is not as straightforward to understand. Are we understanding your point correctly? If so, we absolutely agree with you.
> > >
> > > However, we wonder if the use of LLMs and our inability to pinpoint exactly what the system is doing behind the scenes justifies the negative score given to our paper. Were there any other concerns or questions you had about our work?

---

> > > > ### Comment · Reviewer_vVdR · 2024-12-02
> > > >
> > > > Thank you for the response and I appreciate the efforts made to address my concerns and the acknowledgment of the challenges involved in mathematically grounding methods that leverage LLMs. While I understand the complexity introduced by the use of LLMs in your pipeline, I maintain that mathematical grounding remains a crucial aspect for enhancing the reliability and trustworthiness of interpretability methods. As the authors noted, established methods like LIME, SHAP, and TCAV, benefit from a mathematically transparent foundation. This transparency allows users to better understand and trust the scores generated, even if the methods involve approximations or assumptions. In contrast, the reliance on LLMs introduces a level of opaqueness that makes it harder to systematically analyze or validate the outputs, which ultimately affects the perceived reliability of the proposed LINT score.
> > > >
> > > > I acknowledge the innovation in applying LLMs to interpret programmatic policies, and I value the inclusion of baselines to contextualize your approach. However, the lack of a clear mathematical foundation for the LINT score limits its generalizability and undermines confidence in its applicability beyond the specific domains explored. This was a significant factor in my evaluation and contributed to the score I assigned.

---

> > > > > ### Author Response · Authors · 2024-12-03
> > > > >
> > > > > Thank you for clarifying your point on mathematical grounding as a means of providing transparency of the interpretability metrics. Although we do not fully understand the computation happening in the LLMs, LINT can be quite transparent. This is because LINT produces a natural language intermediate output that we can inspect. Consider, for example, the following excerpt from an explanation LINT generates for an uninterpretable program:
> > > > >
> > > > >
> > > > > *Imagine you have a list of items. Initially, all items are marked as "normal." The program goes through the following steps: It starts at the end of the list and works its way to the beginning. For each item, the program does two actions: First, it keeps the item as "normal" and then displays all the items marked as "greater than normal." [...]*
> > > > >
> > > > >
> > > > > We can quickly see that this description does not make sense. For example, it is not clear what “normal” or “greater than normal” means. This text points to a program for which the LLM failed to generate a sensible explanation and explains the score of 0.0 it assigned to this program. We provide more examples on page 18 of our submission.
> > > > >
> > > > >
> > > > > The natural language bottleneck LINT uses is an (abstract) approximation of the program LINT evaluates, similarly to how the linear functions LIME uses is an approximation of the function it evaluates. Both approximations can be inspected and interpreted by humans.

---

> > > > > > ### Comment · Reviewer_vVdR · 2024-12-03
> > > > > >
> > > > > > Thank you for the clarification and providing the example to clearly explain how LINT can also be interpretable with the help of generated natural language description. Having said that, I still feel this is not good enough to completely rely on as unlike mathematical functions, LLMs can sometimes generate arbitrary text.
> > > > > >
> > > > > > And, based on our discussion I would like to keep my scores.

---

> > > > > > > ### Author Response · Authors · 2024-12-03
> > > > > > >
> > > > > > > Thank you for giving us the chance to clarify this. In our paper, we acknowledge that LLMs can sometimes generate arbitrary text, due to their stochastic nature. The solution is simple: we consider multiple runs of the system when computing the interpretability score (last paragraph of Section 3). Importantly, all these runs can be inspected, as per our discussion.

---

> > > > > > > ### Author Response · Authors · 2024-12-04
> > > > > > >
> > > > > > > We would like to ask the reviewer to consider our contribution in light of the existing alternative approaches to measuring interpretability of programmatic policies. Currently, the approach used in the literature is to make anecdotal claims of interpretability or to run expensive and complicated user studies. LINT is the only inexpensive and reliable metric we have. The metrics you mentioned (LIME, SHAP, and TCAV) have wonderful properties, but they cannot be used with programs.

---

### Author Response · Authors · 2024-11-17
**Summary of Revisions**

We thank the reviewers for their feedback on our work. We have revised the paper according to their feedback and have uploaded it on OpenReview. You will see the changes in blue in the revised manuscript. In order to fit all the changes in the main text, we moved the representative reconstruction example we had in the main text to Appendix C of the revised paper.

We believe the paper has improved substantially after dealing with the questions and suggestions of the reviewers. In particular, the addition of baselines certainly strengthened the contributions of our work. Please do not hesitate to ask further clarifying questions or suggest further changes to our paper.

In addition to clarifying several aspects around the user study (please see the individual answers below), we made the following changes to the paper.

- Added two baselines to our experiments. In one of the baselines, we use LLMs to assign an interpretability score to the programs we evaluate. However, in contrast with LINT, this LLM baseline does not use LINT's natural language bottleneck (see Section 4 of the revised version). We also added a comparison of LINT with simple structural metrics such as number of lines of code, number of loops and conditionals (see Appendix E of the revised version). None of these baselines correlate as strongly as the LINT scores to the I and V-Scores collected in our user study. See Section 5.2 and Appendix E of the revised paper for a longer discussion of these results (**Reviewers vVdR, tPe6, gZii, and deQy**).
- We added in Appendix E empirical results on C programs from an obfuscation competition. (**Reviewers vVdR, tPe6, n7Uq and deQy**)
- Added a brief introduction to programmatic policies at the end of Section 2 (**Reviewer vVdR**).
- Appendix L has been added, containing four tables that detail the statistical differences in the I-Score and V-Score columns, providing clarification on the superscripts used in Tables 4 and 5 for both Karel and MicroRTS (**Reviewer vVdR**).
- Answered all questions the reviewers asked (please see **individual comments** below).

---

### Meta-Review · Area_Chair_ikJP · 2024-12-21

**Metareview:**

The paper introduces the LINT (LLM-based INTerpretability) score, a novel metric for evaluating the interpretability of programmatic policies in reinforcement learning. The methodology involves using large language models (LLMs) in three roles: (1) generates a natural language description of the policy, (2) reconstructs the policy from the explanation, and (3) ensures the explanation doesn't leak unnecessary details for reconstruction. The LINT score quantifies interpretability based on the similarity between the original and reconstructed policies. The method is tested in two domains—MicroRTS and Karel the Robot—using obfuscation experiments and user studies to validate its correlation with human interpretability assessments.

Reasons to accept
- The use of LLMs to quantify interpretability in programmatic policies is interesting and creative.
- The proposed pipeline (explainer, reconstructor, verifier) is well-structured, with explicit design choices and motivations for each step.
- Testing across two domains (MicroRTS and Karel) with obfuscation experiments and user studies demonstrates thoughtful experimental design.
- The authors are transparent about the limitations of their approach, including the stochastic nature of LLMs and the domain-specificity of their experiments.


Reasons to reject
- The paper does not convincingly establish why an interpretability score is necessary for programmatic policies, which are often inherently interpretable by examining their code.
- The methodology does not provide formal guarantees that the LINT score accurately reflects policy interpretability.
- The paper does not benchmark LINT against existing interpretability metrics or heuristic-based methods. This omission raises doubts about whether LINT offers any improvement over simpler approaches.
- The experiments are confined to two relatively simple domains, MicroRTS and Karel, with small sample sizes in user studies. This limits the generalizability of the findings to more complex or real-world scenarios.
- The LINT score (e.g., action and return components) is difficult to interpret, especially without normalization or domain-independent scaling.
- The stochastic nature of LLMs can introduce variability in results, complicating reproducibility. Also, the black-box nature of LLMs undermines the interpretability of the LINT score itself.
- The reliance on multiple LLMs (explainer, reconstructor, verifier) can be computationally expensive, challenging the claim of cost-effectiveness compared to user studies.

While this paper studies a promising research direction (the interpretability of programmatic RL policies) and proposes an interesting approach, I believe its weaknesses outweigh its strengths. Consequently, I recommend rejecting the paper.

**Additional Comments On Reviewer Discussion:**

During the rebuttal period, all five reviewers acknowledged the author's rebuttal, and two reviewers raised their scores.

---

### Decision · Program_Chairs · 2025-01-22

Reject